

# Dry season aerosol iron solubility in tropical northern Australia

V.H.L. Winton[1,a], R. Edwards[1], A.R. Bowie[2,3], M. Keywood[4], A.G. Williams[5], S. Chambers[5], P.W. Selleck[4], M. Desservettaz[6], M. Mallet[7], C. Paton-Walsh[6]

[1]Physics and Astronomy, Curtin University, Perth, Western Australia, Australia
[2]Antarctic Climate and Ecosystems CRC, University of Tasmania, Hobart, Tasmania, Australia
[3]Institute for Marine and Antarctic Studies, University of Tasmania, Hobart, Tasmania, Australia
[4]CSIRO, Ocean and Atmosphere, Aspendale, Victoria
[5]ANSTO, Institute for Environmental Research, Lucas Heights, New South Wales, Australia
[6]Centre for Atmospheric Chemistry, University of Wollongong, Wollongong, New South Wales, Australia
[7]Department of Chemistry, Physics and Mechanical Engineering, Queensland University of Technology, Queensland, Brisbane, 4000, Australia
[a]now at: British Antarctic Survey, Cambridge, United Kingdom

*Correspondence to*: V.H.L Winton (vicwin@bas.ac.uk)

**Abstract.** Marine nitrogen fixation is co-limited by the supply of iron and phosphorus in large areas of the global ocean. Up to 75 % of marine nitrogen fixation may be limited by iron supply due to the relatively high iron requirements of planktonic diazotrophs (Berman-Frank et al., 2001). The deposition of soluble aerosol iron can initiate nitrogen fixation and trigger toxic algal blooms in nitrate-poor tropical waters. There is a large variability in estimates of soluble iron, related to the mixing of aerosol iron sources. Most studies assume that mineral dust represents the primary source of soluble iron in the atmosphere. However, seasonal biomass burning in tropical regions is a potential source of aerosol iron that could explain the large variability of soluble iron in those regions. To investigate aerosol iron sources to the adjacent tropical waters of Australia, the fractional solubility of aerosol iron was determined during the Savannah Fires in the Early Dry Season (SAFIRED) campaign at Gunn Point, Northern Territory, Australia during the dry season in 2014. The source of particulate matter less than 10 μm (PM10) aerosol iron was a mixture of mineral dust, fresh biomass burning aerosol, sea spray and anthropogenic pollution. The mean soluble and total aerosol iron concentrations were 40 and 500 ng m$^{-3}$ respectively. Fractional Fe solubility was relativity high for the majority of the campaign and averaged 8 % but dropped to 3 % during the largest and most proximal fire event. Fractional Fe solubility and proxies for biomass burning (elemental carbon, levoglucosan, oxalate and carbon monoxide) were unrelated throughout the campaign. An explanation of the lack of correlation between fractional Fe solubility and elemental carbon at the biomass burning source is due to the physical properties of elemental carbon, i.e., fresh elemental carbon aerosols are initially hydrophobic, however they can disperse in water after aging and coating with water soluble species in the atmosphere. Combustion aerosols are thought to have a high factional Fe solubility, which can increase during atmospheric transport from the source. Although, biomass burning derived particles may not be a direct source of soluble iron, they can act indirectly as a surface for aerosols iron to bind during atmospheric transport and subsequently be released to the ocean upon deposition. In addition, biomass burning derived aerosols can indirectly impact the fractional solubility of mineral dust. Fractional Fe solubility was highest during dust events at Gunn Point, and could have been enhanced by mixing with biomass burning derived aerosols. Iron in dust may be more soluble in the tropics compared to higher latitudes due to the presence



higher concentrations of biomass burring derived reactive organic species in the atmosphere, such as oxalate, and their potential to enhance the fractional Fe solubility of mineral dust. As the aerosol loading is dominated by biomass burning emissions over the tropical waters in the dry season, additions of biomass burning derived soluble iron could have harmful consequences for initiating nitrogen fixing toxic algal blooms. Future research is required to quantify biomass burning derived particle sources

of soluble iron over tropical waters.

**Key words**

Iron, dust, biomass burning, soluble iron, aerosol, Australia

**1 Introduction**

The deposition and dissolution of aerosols containing trace metals, such as iron (Fe), into the ocean may provide important

micronutrients required for marine primary production in waters where they are depleted, such as the Southern Ocean (e.g. Boyd et al., 2000). Conversely, the deposition of soluble iron can trigger toxic algal blooms in nutrient-poor tropical waters (LaRoche and Breitbarth, 2005). In these waters, iron availability is a primary factor limiting nitrogen fixation (Rueter, 1988;Rueter et al., 1992;Falkowski, 1997;Knapp et al., 2016;Garcia et al., 2015), and the addition of bioavailable iron can influence inputs of newly fixed nitrogen into surface waters (Paerl et al., 1987;Rueter, 1988). A large fraction of nitrogen

fixation is attributed to the filamentous, nonheterocystous cyanobacteria *Trichodesmium* (Paerl et al., 1994;Berman-Frank et al., 2001). Nitrogen fixation has a higher iron requirement, reflected by higher Fe:C quotas of *Trichodesmium* that range between 180–214 µmol Fe mol$^{-1}$ C [*Berman-Frank et al.*, 2001], compared with other marine phytoplankton such as diatoms which have Fe:C quotas ranging between 1 and 7 µmol mol$^{-1}$ (Kustka et al., 2003;Johnson et al., 1997;Rueter et al., 1992). Previous studies have investigated iron availability as a limiting factor to the oceans surrounding Australia. High iron quotas

and high nitrogen fixation rates of *Trichodesmium* were observed during bloom conditions from coastal waters north of Australia (Berman-Frank et al., 2001). Berman-Frank et al. (2001) calculated the potential of nitrogen fixation by *Trichodesmium* in the global ocean, and suggested that nitrogen fixation is iron-limited in around 75 % of the world oceans, assuming present aerosol iron fluxes and sea surface temperatures. In the oligotrophic waters of the north Tasman Sea, southwest Pacific, a 10-fold increase in nitrogen fixation was observed in response to a cyclone which stimulated diazotrophy

by enhanced phosphate availability in the absence of nitrate, and increased dissolved iron supply by wet deposition of Australian dust (Law et al., 2011).

Aerosols are an important source of new iron to the global ocean (Boyd and Ellwood, 2010;Rubin et al., 2011). In tropical waters, several studies have shown that large toxic algal blooms such as dinoflagellate Gymnodinium and cyanobacteria have

30 been stimulated by an atmospheric deposition of nutrients. For example, the deposition of volcanic ash and Saharan dust may





have alleviated the iron limitation of toxic diazotrophic cyanophytes, thus fuelling nitrogen fixation of red tides in the eastern Gulf of Mexico (Lenes et al., 2008;Walsh and Steidinger, 2001). In addition to mineral dust, biomass burning could be a source of bioavailable iron supply fuelling toxic algal blooms in tropical waters. The mortality of a coral reef in 1997 in western Sumatra, Indonesia was linked to iron-fertilisation by Indonesian wildfires causing a giant red tide of *Trichodesmium* (Abram

et al., 2003).

Australia is the primary source of atmospheric dust in the Southern Hemisphere, episodically supplying dust to the Tasman Sea and Southern Ocean (McTainsh et al., 2005). There are two general dust pathways from the Australian continent to the ocean (Bowler, 1976;Sprigg, 1982). First, the transport of dust from the southeast aeolian dust path is associated with easterly-

moving frontal systems within the zonal westerly winds, and its deposition has been observed in sediment in the southwest Pacific Ocean and glaciers in New Zealand (Hesse and McTainsh, 2003;Johnston, 2001;Hesse, 1994;Marx et al., 2005). Second, the transport of dust from the northwest aeolian dust path is associated with the easterly trade winds. In spite of the ubiquity of aeolian dust in Australia and surrounding oceans, there are few studies investigating the composition of the dust on a continental scale. Some, but not all, Australian dust storms are thought to stimulate phytoplankton blooms (Shaw et al.,

2008;Cropp et al., 2013;Gabric et al., 2010;Mackie et al., 2008). Soluble iron deposition models indicate that dust and combustion aerosols from the Australian continent can be deposited in other coastal regions of Australia, such as the Indian Ocean and waters north of Australia (Ito, 2015;Mahowald et al., 2005). Studies from ice cores show that Australian dust and refractory black carbon (rBC), a proxy for biomass burning, can be transported long-range to Antarctica (Revel-Rolland et al., 2006;De Deckker et al., 2010;Bisiaux et al., 2012).

We note that the difference in the terminology of black carbon versus elemental carbon in the literature reflects the characteristics of carbonaceous matter and the analytical method employed (Andreae and Gelencsér, 2006;Petzold et al., 2013;Bond et al., 2013). The terminology used here follows that recommended by Petzold et al. (2013), that is, the term "black carbon" is used qualitatively when referring to material that shares some of the characteristics of black carbon (carbonaceous

composition combined with light-absorbing properties); "elemental carbon" refers to data derived from methods that are specific to the carbon content of carbonaceous matter using thermal desorption methods; "refractory black carbon" (rBC) is used instead of black carbon for measurements derived from incandescence methods; while mixed particles containing black carbon are referred to as "black carbon-containing particles." The data reported in this study was analysed using thermal desorption methods and is referred to as elemental carbon.

The fractional solubility of aerosol iron is an important variable determining iron availability for biological uptake. On a global scale, the large variability in the observed fractional iron solubility results, in part, from a mixture of different aerosol sources. To date most studies have assumed that mineral dust aerosols represent the primary source of soluble iron in the atmosphere (e.g. Baker and Croot, 2010). Mineral dust has a low fractional iron solubility (~0.5-2 %), whereas the presence of other soluble





iron sources, such as those originating from biomass burning and oil combustion have a higher fractional iron solubility (Chuang et al., 2005;Guieu et al., 2005;Sedwick et al., 2007;Schroth et al., 2009;Ito, 2011). Although, the total iron content of fresh smoke is small 0.01-1.2 % (Maenhaut et al., 2002;Yamasoe et al., 2000;Reid et al., 2005), estimates of fractional iron solubility from fire combustion are large (1 to 60 %). This large variability may be related to characteristics of the biomass, and fire as well as that of the underlying terrain (Ito, 2011;Paris et al., 2010).

An investigation of rBC deposition to the Ross Sea, Antarctica found that variability in annual deposition parallels austral dry season rBC emissions, dust and soluble iron deposition (Winton et al. (2016b). Iron associated with biomass burning may be an important source of iron to the Southern Ocean. Austral dry season rBC emissions in the Southern primarily occur in the intertropical convergence zone (ITCZ) of Africa, Australia and South America (Giglio et al., 2013). Australia is the closest of these regions to the Ross Sea. Here, biomass burning constitutes a large source of annual dry season aerosol emissions over northern and central Australia, and episodic austral summer wild fire in southern and eastern Australia (e.g. Meyer et al., 2008). To date, data for Australian aerosol soluble iron sources is sparse (e.g. Mackie et al., 2008;Mackie et al., 2005), and no soluble aerosol iron data exists for Australian fire sources. The dry season tropical savannah burning region in northern Australian provides an ideal location to further investigate biomass burning derived fractional iron solubility at the source. This study reports fractional iron solubility as estimates of mixed dust and fresh biomass burning (black carbon-containing particles) derived iron input to north Australian waters, as part of the Savannah Fires in the Early Dry Season (SAFIRED) campaign from the Australian Tropical Atmospheric Research Station (ATARS), Gunn Point, Northern Territory, Australia.

## 2. Methods

### 2.1. Study site

The CSIRO Oceans and Atmosphere and the Australian Bureau of Meteorology has established a station at the ATARS, Gunn Point, Northern Territory, Australia (12°14'56.6"S, 131°02'40.8"E) (Fig. 1) monitoring meteorological parameters since 2010. The SAFIRED campaign occurred at the ATARS throughout June 2014 to investigate the chemical and physical properties of aerosols and gases associated in fresh and aged smoke plumes from biomass burning. A range of atmospheric species including greenhouse gases, aerosols, and additional meteorological parameters were monitored throughout the campaign. An overview of the campaign and description of the Gunn Point study site can be found in *Mallet et al.* [in prep]. The Northern Territory of Australia experiences hot, humid and wet conditions during the summer months of December through to March, and dry conditions for the rest of the year. The savannah region gives rise to frequent fires between June and November. Indonesia and the tropical Timor Sea lie to the north of Gunn Point. The large desert regions of central Australia lie to the south. During the sampling campaign, strong winds from the southeast brought continental atmospheric conditions to the site (Fig. 1).



## 2.2. Sampling and blanks

Daily aerosol filters were collected using two Volumetric Flow Controlled (VFC) high-volume aerosol samplers (Ecotech) attached with particulate matter less than 10 µm (PM10) size selective inlets (Ecotech). The samplers were operated simultaneously. The first high-volume sampler was used to collect aerosols on acid cleaned cellulose Whatman 41 filter sheets

(20 x 25 cm) to determine the soluble and total fraction of aerosol iron and other trace elements. The second sampler was used to collect aerosol samples on quartz filters for elemental carbon and major cation and anion measurements. The aerosol samplers were located on the roof of a shipping container (at a height of 5 m above ground level (agl)) at the ATARS. A total of 23 trace metal aerosol filters were sampled daily between 4 and 27 June 2014 and are described in Table 1.

All sample preparation and analysis was conducted in a trace metal clean laboratory (class 100 metal-free environment with high-efficiency particulate arrestance (HEPA) filtered air) at Curtin University following the ultra-clean methodology. Details for ultra-pure reagents, ultra-pure water, acid-washing of apparatus and Whatman 41 filters and filter sub-sampling are described in Winton et al. (2016a).

Loading and changing of aerosol collection substrates was carried out in a designated clean area at the ATARS. Aerosol laden filters were transferred into the individual pre-acid-washed zip-lock plastic bags immediately after collection and stored frozen until analysis at Curtin University. Three types of filter blanks were carried out; i) laboratory filter blanks (n=6) (acid-washed Whatman 41 filter papers that underwent the laboratory procedures without going into the field), ii) procedural filter blanks (n=4) (filters that had been treated as for normal samples i.e. acid-washed, but which were not otherwise used. Once a week,

during daily filter change over, a procedural blank filter was mounted in the aerosol collector for 10 minutes without the collector pump in operation to give an indication of the operational blank associated with the sampling procedure), and iii) 24 hour exposure filter blanks sampled at the beginning and end of the field campaign (n=2) (filters treated like a procedural blank, but left it in the collector for 24 hours without switching the collector on).

## 2.3. Trace element analysis

### 2.3.1. Water solubility of aerosol iron

Soluble elements were extracted from aerosol filters (GP1-23), exposure blanks (EB1-2), and procedural blanks (PB1-4). Soluble elements (Al, Ti, V, Mn, Fe, Pb) were extracted from the filter using an instantaneous flow-through water leach consisting of 20 separate passes of 50 mL of ultra-pure water. A total volume of 1000 mL was filtered through each aerosol filter. Only leaches 1-4, 6, 10 and 20 were collected and analysed, following a test where we determined the maximum number

of leaches required to leach the majority of soluble iron from the aerosols. In this test, we leached sample GP5 with 10 passes of 50 mL ultra-pure water. The experiment was repeated three times using three separate punches from the same sample. Leachates were collected in 50 mL acid-washed polypropylene (PP) Corning® tubes and acidified to 1 % ultra-pure HCl.





Leaches 5, 7-8 and 11-19 were estimated by fitting a power law curve to the each sample. Total water soluble trace elements were calculated by summing the soluble iron extracted from 20 leaches. The leachates from sample GP22 were filtered through 0.2 µm acid-cleaned Polyvinylidene Fluoride (PVDF) syringe filters. Blank concentrations for exposure blank and procedural blank filters averaged ~0.4 ng g$^{-1}$ and 0.5 ng g$^{-1}$ of soluble Fe respectively.

### 2.3.2. Total iron concentrations

Total PM10 trace element (Na, Al, K, Ti, V, Cr, Mn, Fe, Mo, Pb, As) concentrations were determined following recommendations from the 2008 GEOTRACES intercalibration experiment for the analysis of marine aerosols (Morton et al., 2013). Details for digestion of filter samples and certified reference materials (MESS-3 marine sediment, National Research Council, Canada, and QC-TMFM-A spiked trace metals (1-10 µg of trace metal per filter) on nitrocellulose filter (TMF), High Purity Standards) can be found in Winton et al. (2016a). Recovery rates for elements are reported in Table S1. Blank concentrations for Savillex® digestion beakers, exposure blank, procedural blank and laboratory blank filters averaged ~4 ng g$^{-1}$ (n=3), ~150 ng g$^{-1}$ (n=2), ~140 ng g$^{-1}$ (n=4), and ~140 ng g$^{-1}$ (n=6) of total iron respectively.

### 2.3.3. High-resolution inductively coupled plasma mass spectrometry analysis

Leachates and resuspended total digests were analysed by high-resolution inductively coupled plasma mass spectrometry (HR-ICP-MS, Element XR ThermoFisher) at Curtin University following Winton et al. (2016a). All samples and standards were prepared on a similar matrix basis. During the course of the sample sequence, an EPA standard (Cat. #ICP-200.7-6 Solution A, High Purity Standards) was measured regularly for quality control (QC). Laboratory vial blanks were also measured regularly during the course of the sample sequence determined by carrying out identical analytical procedures as the leachates but vials were filled with ultra-pure water rather than the leachate. Leachate concentrations were corrected for laboratory blanks, and the total digest concentrations corrected for the Savillex® digestion beaker blanks. Measured elements and the spectral resolutions, along with typical operating conditions, are reported in Table S2.

### 2.4. Air mass back trajectories

To identify the source region of air masses arriving at Gunn Point, we use hourly 5-day (120-hr) back trajectories (ending at 12 m agl) to reconstruct atmospheric circulation leading up to, and during, individual events. Air mass back trajectories were produced using the 'Trajectory program' based on the NOAA ARL HYbrid Single-Particle Lagrangian Integrated Trajectory (HYsplit) model (Draxler and Rolph, 2003) (http://ready.arl.noaa.gov/HYSPLIT.php). Global Data Assimilation System model files (GDAS) obtained from NOAA ARL FTP (http://ready.arl.noaa.gov/gdas1.php) were used to drive the model. The resolution used was 0.5 degrees.



### 2.5. Radon concentrations

Natural radioactive noble gas radon-222 (radon) is a useful quantitative indicator of diurnal to synoptic scale mixing processes within the continental lower troposphere (Chambers et al., 2015), and a suitable tracer for continental air (e.g. Dörr et al., 1983). This is because radon has a half-life of 3.8 days and only one source and sink i.e. radon is solely produced from terrestrial surfaces and is lost from the atmosphere by radioactive decay. The ANSTO-built 700 L dual-flow-loop two-filter radon detector (e.g. Whittlestone and Zahorowski, 1998;Chambers et al., 2014) samples air from 12 m above ground level (a.g.l.). A coarse aerosol filter and dehumidifier are installed "upstream" of the detector, as well as a 400 L delay volume to ensure that thoron ($^{220}$Rn, half-life 55 s) concentrations in the inlet air stream are reduced to less than 0.5 % of their ambient values. The detector's response time is around 45 minutes, and its lower limit of determination is 40 - 50 mBq m$^{-3}$. Calibrations are performed on a monthly basis by injecting radon from a PYLON 101.15 ± 4 % kBq Ra-226 source (12.745 Bq min$^{-1}$ $^{222}$Rn), traceable to NIST standards, and instrumental background is checked every 3 months. The time series for the Gunn Point campaign duration has been separated into contributions from a) advective (reflects the air mass fetch history over the last two weeks), and b) diurnal variability (indication of the change in mixing depth) following Chambers et al. (2015). The advective and diurnal components of the radon time series were integrated to daily values based on a noon-to-noon integration window.

### 2.6. Major cation and anion

Major cation and anion measurements, including oxalate and levoglucosan, were performed on the PM10 quartz filters at CSIRO Ocean and Atmosphere. The filters were Pall tissuquartz filters p/n 7204, baked them at 40 °C for 4-8 hours before use and stored frozen after sampling until analysis. A 6.25 cm$^2$ section was cut out of the filter and extracted in 5 mL of ultra-pure water. The quartz filter extracts were analysed for major water soluble ions by suppressed ion chromatography (IC) and for anhydrous sugars including levoglucosan by high-performance anion-exchange chromatography with pulsed amperometric detection (HPAEC-PAD) (Iinuma et al., 2009;Hibberd et al., 2013). Three exposure blanks were analysed alongside the sample filters. All concentrations were blank corrected.

### 2.7. Elemental carbon

Elemental carbon analysis was performed on the PM10 quartz filters at CSIRO Ocean and Atmosphere using a DRI Model 2001A Thermal-Optical Carbon Analyser following the IMPROVE-A temperature protocol (Chow et al., 2007).

### 3. Results

### 3.1. Air mass back trajectories

We identified predominate fetch regions that gave rise to particular events; a) 'inland, low population density,' b) 'coastal, moderate urban/industrial activity,' c) 'major urban/industrial activity,' and d) 'southerly.' No trajectories fell in the 'mainly





oceanic' category. The mean trajectory for each group was calculated and these representative trajectories are indicated in Fig. 1.

## 3.2. Biomass burning events during the campaign

A number of biomass burning markers were used to identify burn events during the campaign, including carbon monoxide

(CO) [*M. Desservettaz, Pers. Comm.*, 2014], levoglucosan [*P. Selleck Pers. Comm.*, 2015], oxalate [*P. Selleck Pers. Comm.*, 2015] and elemental carbon [*P. Selleck Pers. Comm.*, 2015]. Nine burn events were identified, with the three major events occurring on the 9 June (fire event "E"), 25 June (fire event "FG") and 26-27 June (fire event "HI) 2014 [*Mallet et al.*, in prep]. Smoke transported to the Gunn Point in fire event "E" was distally sourced i.e. >400 km, while smoke from the largest fire events "FG" and "HI" was proximal i.e. within 15 km of the ATARS (fire locations from Sentinel Hotspot (MODIS and VIIRS

satellites); *Mallet et al.* [in prep]). Due to the different time periods the CO data (minute) and aerosol samples (daily) integrate, we have grouped fire events "F" and "G" into one event (fire event "FG") and "H" and "I" into another event (fire event "HI") for this study. C3 tree burning was identified from $\delta CO_2$ during the 25 June burning event [*C. Paton-Walsh, Pers. Comm.*, 2014]. In addition, concentrations of non-sea salt-potassium (nss-K) and nss-K normalised relative to other elemental concentration are good indicators of biomass burning (Yamasoe et al., 2000;Maenhaut et al., 2002). There is good agreement

between nss-K concentrations and carbon monoxide (CO) ($r^2$=0.55) (Fig. S1). There is also a good correlation between nss-K and elemental carbon concentrations ($r^2$=0.74) (Fig. S1). Following Guieu et al. (2005), we normalise nss-K to Fe and compare the ratio to crustal aerosols (nss-K/Fe=0.49; Wedepohl (1995)) and polluted aerosols (nss-K/Fe=0.27; urban particulate patter – standard reference #1648, NIST). During the campaign the nss-K/Fe ratio ranged between 0.2-4.9. This relativity large range was similar to nss-K/Fe of 0.63-1.8 reported by Guieu et al. (2005) for Mediterranean summer fires in 2003. They suggested

the higher ratio reflected a primarily fire origin for nss-K. The nss-K/Fe ratio was especially high during the three fire events in this study (nss-K/Fe ~4) and during the high elemental carbon concentrations between 21-27 June (nss-K/Fe~2.2), compared to non-fire events (nss-K/Fe ~1.3).

## 3.3. Sequential iron leaching

To determine the number of ultra-pure water leaches required to leach the majority of water soluble Fe from the Gunn Point

aerosol filters, we conducted a test on the GP5 sample where three filter aliquots were leached with 10 sequential passes of 50 mL of water. The leaching results in Fig. S2 show that there is an exponential decrease in soluble Fe extracted with successive leaches. Even after 1 L of ultra-pure water was passed through the aerosol filter, soluble Fe was still being extracted. We therefore leached each sample with 20 passes of ultra-pure water totalling 1 L. We acknowledge that beyond 20 leaches, soluble Fe could continue to be leached out of the aerosol filter and thus the data reported here reflect "instantaneous" soluble

Fe and may underestimate the total concentration of soluble Fe over greater leach volumes. Figure 2 highlights the percentage of soluble Fe in each leach. Around half of the soluble Fe is extracted in the first leach. The leachates from GP22 were re-filtered through 0.2 µm membranes to calculate the <0.2 µm soluble Fe fraction of the total soluble Fe. The concentration of



soluble Fe in the <0.2 µm fraction of the leachate contained 70 % of the total soluble Fe in GP22 and follows the same exponential decreases with successive leaches. Leachates showed a large variability in their colour, and visual observation after particle settling showed that they contained a large number of very fine particles even in the <0.2 µm fraction.

### 3.4. Aerosol iron mass concentrations

Total and soluble PM10 trace element concentrations are illustrated in Figs. 3-5. Total PM10 aerosol Fe ranges from 60-1164 ng m$^{-3}$, while soluble aerosol Fe ranges from 7-141 ng m$^{-3}$. A strong linear correlation between in the soluble trace elements of Fe, Al, and Ti was found during the campaign (Fig. 5), i.e., soluble Fe and soluble Al ($r^2$=0.97), and soluble Fe and soluble Ti ($r^2$=0.86). Likewise, a strong linear correlation was found between the total PM10 trace elements of Fe, Al, V and Ti, i.e., total Fe and total Al ($r^2$=0.96), total Fe and total Ti ($r^2$=0.98), and total Fe and total V ($r^2$=0.99) (Fig. 3). There is considerable

temporal variability in water soluble and total trace elements throughout the campaign. Two distinct events in soluble Al, Ti, and Fe and total Al, Ti, V, and Fe are observed on the 7-8 June 2014 (e.g. GP4-5; 78-141 ng m$^{-3}$ of soluble Fe and 1158-1164 ng m$^{-3}$ of total Fe), and 14-15 June 2014 (e.g. GP11-12; 52-92 ng m$^{-3}$ of Fe and 677-928 ng m$^{-3}$ of total Fe) (Figs. 4 and 6). On the 20 June 2014 high concentrations of these crustal like elements (total and soluble Al, Ti, V, Fe) were also observed (e.g. GP17; 67 ng m$^{-3}$ of soluble Fe and 1129 ng m$^{-3}$ of total Fe) and are associated with marine air mass conditions. In addition,

total Cr, As, Mo, V, and Na also peaked during the marine conditions on the 20 June (Fig. 5). This same set of total trace elements (Cr, As, Mo, and V) also peaked on the 9 June during fire event "E" (Fig. 5). During the three fire events "E", "FG" and "HI" on the 9, 25 and 26-27 June, daily integrated CO and nss-K were high at the same time as total As, Mn, Pb, and soluble V, Mn, Pb (Fig. 4).

### 3.5. Dry deposition iron flux

Estimates of the Fe dry deposition rate ($F_{dry}$) to adjacent north Australian surface waters were calculated using Eq. (1) from the total and soluble Fe concentrations ($C_{aerosol}$) using a dry deposition velocity ($V_{dry}$) of 2 cm s$^{-1}$ for coastal areas (Duce et al., 1991), following Baker et al. (2003) for aerosols collected along a transect in the Atlantic Ocean and Baker et al. (2007) for the atmosphere over the tropical Atlantic Ocean. Estimates of Fe dry deposition rates for Gunn Point aerosols are reported in Table 2.

$$F_{dry} = C_{aerosol} \; x \; V_{dry \; deposition} \tag{1}$$

### 4. Discussion

### 4.1. Iron mass concentrations

Total PM10 Fe concentrations during the fire events (0.06-1.2 µg m$^3$; Fig. 3) are in good agreement with fire aerosol studies reported from the Alta Floresta, Amazon forest, Brazil between 1996-1998, which range from 0.3 to 1.2 µg m$^{-3}$ of total Fe





(Maenhaut et al., 2002). The soluble Fe concentrations at Gunn Point (7 to 141 ng m³) are less than soluble Fe concentrations reported in smoke from the Mediterranean and West African Sahel (350 ng m³; Guieu et al. (2005)) and (130 ng m³; Paris et al. (2010)) respectively. There is sparse total and soluble aerosol Fe concentration data from Australian aerosols, however our estimates agree with reported total Fe concentrations at Jabiru, northern Australia (148 ng m$^{-3}$), at Sydney, New South Wales

(150 ng m$^{-3}$) and modelled values for the north Australia coastal area (200-1000 ng m$^{-3}$) (Mahowald et al., 2009; and references within). Compared to aerosol Fe from other tropical regions, our estimates at the Australian source are greater than aerosol Fe concentrations reported for the tropical Atlantic open ocean (total Fe 6-56 ng m$^{-3}$ and soluble Fe 6-28 ng m$^{-3}$) (Baker et al., 2006).

## 4.2. Dry deposition estimates of soluble and total iron

Estimates of soluble and total PM10 Fe dry deposition fluxes are reported in Table 2 and range from 0.2 ± 0.1 to 4 ± 2 µmol m$^{-2}$ d$^{-1}$ for soluble Fe and 2 ± 1 to 36 ± 18 µmol m$^{-2}$ d$^{-1}$ for total PM10 Fe. The dry deposition Fe estimates for air over Gunn Point are compared to other aerosol studies from the Southern Hemisphere and topical regions. The mean soluble Fe dry deposition flux (~1 µmol m$^{-2}$ d$^{-1}$) is greater than soluble Fe dry deposition fluxes over open ocean tropical Atlantic waters (0.01-0.1 µmol m$^{-2}$ d$^{-1}$) (Baker et al., 2003). The estimates at Gunn Point are also higher than those reported for the Southern

Ocean, for example, 2-7 nmol m$^{-2}$ d$^{-1}$ reported by Bowie et al. (2009), 0.04-3 nmol m$^{-2}$ d$^{-1}$ reported by Wagener et al. (2008), 2-7 nmol m$^{-2}$ d$^{-1}$ reported by Baker et al. (2013) for the South Atlantic, and 0.1-7 nmol m$^{-2}$ d$^{-1}$ for baseline air over the Southern Ocean reported by Winton et al. (2015). Estimates of total PM10 Fe dry deposition fluxes at Gunn Point are higher than those for the Southern Ocean (0.8-120 nmol m$^{-2}$ d$^{-1}$) (Winton et al., 2015) and the South Atlantic (100–300 nmol m$^{-2}$ d$^{-1}$) (Baker et al., 2013). The Gunn Point data are in good agreement with estimates from global models for the north Australia coastal region.

Mahowald et al. (2009) estimates total Fe deposition in the north Australia to be 0.04-1 g m$^{-2}$ y$^{-1}$, and our mean total PM10 Fe flux of 0.3 g m$^{-2}$ y$^{-1}$ falls within this range.

## 4.3. Fractional iron solubility

The fractional Fe solubility ranged from 2 % to 12 % during the study (Figs. 6 and 7). These estimates are within the 0.6 to 40 % range of fractional Fe solubility reported for biomass burning (Paris et al., 2010;Bowie et al., 2009;Ito, 2011;Ito, 2015;Guieu

et al., 2005). Fractional Fe solubility was highest between 4 and 20 June with a mean fractional Fe solubility of 10 ± 2 %. After 20 June, the fractional Fe solubility dropped to 3 ± 1 % between the 21 and 27 June when fires were proximal to the ATARS. These estimates are similar to dust estimates around 0.5-2 % at relatively high Fe mass concentrations. During dust event "A" fractional Fe solubility peaked at ~12 %, fractional Al solubility at ~14 %, and fractional Ti solubility at ~5 %. Fractional Al and Ti solubility also decreased on 24 June from 10 ± 2 % to 3 ± 1 % for Al and from 6 ± 2 % to 2 ± 1 % for Ti. Fractional

Mn solubility was relatively constant throughout the campaign and averaged ~72 % ± 11 %. Fractional Mn solubility was highest during fire events "FG" and "HI" when the fractional solubility of Ti and Al was low.





A synthesis of global aerosol Fe solubility data sets complied by Sholkovitz et al. (2012) displayed an inverse hyperbolic relationship between the total Fe concentration and fractional Fe solubility (Fig. 7). This characteristic relationship is common over large regions of the global ocean, and has been attributed to the mixing of mineral dust with a low Fe solubility and other soluble Fe aerosols from combustion sources (e.g. Sedwick et al., 2007). We have plotted the Gunn Point aerosol Fe data with

the Southern Hemisphere compilation in Fig. 7. Our data sits within two narrow clusters i) moderate fractional Fe solubility and moderate total PM10 aerosol Fe loading between the 4 and 19 June and ii) low fractional Fe solubility and low total PM10 aerosol Fe loading between the 24 and 26 June (highlighted in Fig. 7).

### 4.4. Enrichment factor analysis

Crustal enrichment factors (EF) were calculated using the Wedepohl (1995) compilation of the continent crust to determine

the contribution of mineral dust to the observed total elemental concentrations. Total Ti was used as a marker for mineral dust. For an element (Z) in a sample, the EF relative to Ti is calculated using Eq. (2).

$$EF = \frac{(Z/Ti)_{sample}}{(Z/Ti)_{crust}}$$  (2)

The enrichment factors of Gunn Point filters are used to evaluate the level of contamination from mineral dust versus other sources. From the EF reported in Table 3, two groups of trace elements were identified. The first was Al and Fe which have

low EFs between 0.6–1.3 throughout the campaign. Enrichment factors in this range are similar to the upper continental crust (i.e., EF between 0.7 and 2), suggesting that these trace metals might have originated from that source. Aerosol Fe has an EF between 0.9 and 1.3 and does not show Fe enrichment, implying that anthropogenic pollution was not a dominant source of Fe to Gunn Point.

The second group of trace elements consists of Cr, Mn, and Pb, and nss-K which had maximum EFs between 5 and 12 (i.e., moderate enrichment >2 EF <10 to enriched EF >10), indicating mixed sources. Cr showed moderate enrichment (EF of ~4) during the marine event, and also on the 13 June (EF of 8) when back trajectories crossed coastal Queensland (i.e., trajectory type "B"). Anthropogenic Cr could be sourced from industry and combustion (Pacyna and Nriagu, 1988).

Manganese, Pb and nss-K were enriched at three times throughout the campaign. Mn enrichment could be associated with particles that became airborne by soil dispersion or anthropogenic emissions (e.g., smelting and unleaded car fuel) (e.g. Parekh, 1990). Atmospheric Pb originates from the geological weathering, smelting, burning of unleaded fuel, and coal combustion (e.g. Bollhöfer et al., 1999;Bollhöfer and Rosman, 2000;Bollhöfer and Rosman, 2001). The air mass back trajectories showed that during the elevated EF of Mn, Pb and nss-K, the wind direction was from the southeast and the fetch area included major

cities and industrial areas in Brisbane (i.e. trajectory types "A" and C" (Fig. 1). Manganese (EF up to 5), Pb (EF between 5 and 12) and nss-K (EF between 3 and 6) showed moderate enrichment during fire events "E", and "FG." Enrichment of these elements also occurred during the 16-18 June when trajectories came from the south and passed over major smelting sites i.e.



Port Pire and Olympic Dam in South Australia and Mt Isa, Queensland (i.e., trajectory type "D"). The enrichment factors and air mass back trajectories suggest that both smelting and biomass burning are sources of Mn, Pb and nss-K aerosols at these three times.

5    Lead can also originate from resuspension of contaminated material. Enrichment factors for Pb during dust events "A" and "B" were similar to the upper crust suggesting a dust source for aerosol Pb at this time. Enrichments factors of V were constant throughout the campaign, and ranged from 1.6 to 3 indicating the moderate EF for V could be related to contamination from both atmospheric mineral dust and fuel oil burning e.g. from ship exhaust (Desboeufs et al., 2005;Jang et al., 2007;Hope, 2008).

Lower enrichment factors have been observed in marine air masses at Cape Verde (collected on a 30 m tower), indicating that the elements were likely of crustal origin. On the other hand, higher EFs have been reported in marine air masses that crossed Europe and North America, indicating that the elements were of anthropogenic origin (Fomba et al., 2012). The low EF in Al and Fe and higher EF in Mn, Cr, nss-K and Pb during fire events and air masses that passed over major smelting sites are likely

to also reflect a combination of crustal, marine and anthropogenic aerosols.

### 4.5. Aerosol sources

Based on a combination of radon, air mass trajectories, enrichment factor analysis and biomass burning markers, trace elements have been grouped into three sources: dust, biomass burning and marine sources.

### 4.5.1. Dust events

Dust events "A" and "B" are characterised by peaks in total PM10 Al, Ti, Fe and V, and correspond to low diurnal radon i.e. when conditions are well mixed (Fig. 3). During these two dust events, back trajectories corresponded to times when the air mass passed over central Australian desert and low population areas of inland Australia, i.e., dust event "A": trajectory type "c," and dust event "B": trajectory type "d," (Fig. 1). Enrichment factors of Al (EF of 0.6-0.7), V (EF of 1.8-2.0) and Fe (EF of 1.0-1.1) during these two dust events suggest that these trace metals have a similar composition to the upper crust (i.e. EF 0.7-2) and thus most likely originate from crustal material. Soluble concentrations of Fe, Al, Ti, Mn and V were high during

dust events "A" and "B". Fractional Fe solubility was highest in dust event "A" (12 %).

### 4.5.2. Anthropogenic and sea spray sources in marine conditions

On the 20 June 2014 marine conditions were identified by a decrease in aerosol particle counts and a change in the particle size distribution from unimodal to bimodal with modes at 20 nm and 70 nm [*Mallet et al.*, in prep]. During these conditions,

total Na peaked to 3.1 µg m$^{-3}$ add the Na/Mg ratio was similar to seawater which is indicative of sea spray (Figs. 4-6). The Na signal was greater during times of low advective radon i.e. when air masses were predominantly spending less time over land



and more time over ocean regions. Total PM10 Al, Ti, Fe, V, as well as total PM10 Cr, Mo and As peaked during these conditions. There was a moderate rise in soluble Al, Ti and Fe. Enrichment factors for Al, V and Fe during marine conditions show that these trace metals have a similar composition to the upper crust (i.e. EF 0.7-2) and might originate from crustal material. Fractional Fe solubility was ~6 % in the marine air mass. During these conditions there were high concentrations of total PM10 Mo, Cr and As (Fig. 4), and the enrichment factor of Cr was moderately enriched (EF of 4.2). Small emissions of these three trace elements come from natural sources such as mineral dust, or volcanic emissions in the case of As. The greatest anthropogenic As emissions come from pyrometallurgical operations in the production of nonferrous metals, combustion of fossil fuels and the use of pesticides. Anthropogenic sources of aerosol Mo include the erosion of vehicle parts (Lane et al., 2013), while anthropogenic Cr in the atmosphere is sourced from metallurgical and alloy steel, chemical, and refractory industry, coal and oil combustion, cement manufacturing, and refuse incineration (Pacyna and Nriagu, 1988). Back trajectories passed over coastal regions of Queensland, i.e., type "b" (Fig. 1), suggesting that air masses could have contained a mixture of dust and anthropogenic derived aerosols.

### 4.5.3. Fire events

The discussion of aerosol Fe during fire events focusses on the three major events occurring on the 9 June (fire event "E"), 25 June (fire event "FG") and 26-27 June (fire event "HI") 2014 [*Mallet et al.*, in prep]. These events were comprised of C3 burning inferred from $\delta^{13}C$ in $CO_2$ during the campaign [*C. Paton-Walsh, Pers. Comm.* [2014], and aerosol mass concentrations were higher than usual for the campaign. Concentrations of non-refractory particulate matter less than 1 µm (PM1) increased up to 500 µg m$^{-3}$ [*Mallet et al.*, in prep]. During these fire events, total PM10 Fe (400-800 ng m$^{-3}$) and soluble Fe (7-66 ng m$^{-3}$) concentrations and fractional Fe solubility (~3 %) were considerably less than the average concentration and fractional solubility throughout the campaign. For the first half of the campaign, from the 4-20 June, fractional Fe solubility averaged 10 % and then dropped to ~3 % between 21-27 June. During the drop in fractional Fe solubility, soluble and total PM10 Fe concentrations were the lowest found throughout the campaign.

We compare the fractional Fe solubility to biomass burring proxies during the campaign. There is no relationship between fractional Fe solubility and the biomass burning tracers (nss-K, elemental carbon, oxalate or levoglucosan; Fig. S2). However, fractional Fe solubility is typically higher when elemental carbon concentrations are low and vice versa, excluding dust event "A" ($r^2$=0.33). During the exceptionally large fire events "FG" and "HI", fractional Fe solubility was the lowest (~3 %). In agreement with high concentrations of elemental carbon, oxalate and levoglucosan at this time, visual examination the aerosol filters showed that between the 21-27 June, filters were exceptionally caked with soot. The lowest fractional Fe solubility during fire event "FG" and "HI" is related to proximal fires, whilst the fractional Fe solubility was relatively high during fire event "E" when fires were distantly sourced.




### 4.6. Biomass burning derived soluble iron – combustion and transport

Biomass burning is a potentially large source of soluble Fe to the open ocean (Guieu et al., 2005;Ito, 2015, 2013). Ito (2015) estimate that in the north Australian coastal region, biomass burning contributes around 60-70 % of the total soluble Fe deposition, where the dust contribution is less (40-50 %). Australia is one of the primary regions of biomass burning in the

5 Southern Hemisphere (Giglio et al., 2013) and constitutes a large source of annual dry season aerosol emissions over northern Australia, and episodic austral summer wildfires in southern and eastern Australia (e.g. Meyer et al., 2008). Modelling by Ito (2015) suggests biomass burning derived soluble Fe contributes substantial inputs of soluble Fe to tropical and Southern Ocean waters downwind of Australia. Over the Southern Ocean, there is a larger contribution of biomass burning aerosol Fe than mineral dust and fossil fuel combustion derived aerosol Fe. Therefore biomass burning could be an important source of soluble

Fe to both tropical and Southern Ocean waters surrounding Australia.

The particle size distribution of biomass burning aerosols was typically centred around 95 nm (with 95 % of all particles between 30-280 nm) for nearby fresh smoke at Gunn Point and around 120 nm (95 % of all particles were between 40-390 nm) for distant aged smoke at Gunn Point [*Mallet et al.*, in prep]. While the literature shows that for aged/long-range

transported smoke, the particle sized distribution ranges between 75-540 nm (Kipling et al., 2013). As the particle size distribution for black carbon is in the nanometre range, any Fe associated with black carbon-containing particles is operationally defined as soluble (i.e. <0.2 µm). The majority of soluble Fe in Gunn Point aerosols was within the <0.2 µm pool (i.e., the concentration of soluble Fe <0.2 µm was 7 ng m$^{-3}$ compared to the bulk soluble Fe concentration of 10 ng m$^{-3}$ in GP22). Leachates showed large variability in their colour, and visual observation after particle settling showed that they

contained a large number of very fine particles in the <0.2 µm filtered fraction. Recent studies show that nanometre-sized Fe particles are potentially bioavailable (e.g. Raiswell et al., 2008), and thus future work should be directed to quantifying biomass burning sources of bioavailable Fe in tropical and remote oceanic regions.

Iron can be bound to long-range transportable black carbon-containing particle aggregates deposited in Antarctic snow and ice

(Ellis et al., 2015). Little is known about the process in which Fe is bound to black carbon-containing particle aggregates, but Fe could originate from Fe contained within the biomass (Maenhaut et al., 2002;Yamasoe et al., 2000;Reid et al., 2005), soil Fe incorporated into the aerosol mixture during combustion, or the binding of aerosol Fe to black carbon-containing particles in the atmosphere during transport. Using single particle analysis, mixed sources (e.g. black carbon and aluminosilicates) were detected in long-range transported aerosols to Antarctica (Ellis et al., 2015). It is not surprising that the fractional Fe solubility

data (2-12 %) plots mid-range for aerosol Fe sources characteristic of dust and combustion in the Southern Hemisphere (Fig. 7).



Generally, the fractional Fe solubility was lower when fresh elemental carbon concentrations were high and sourced from proximal fires at Gunn Point (Fig. 6). This relationship could be related to the hydrophobic, i.e. water insoluble, nature of black carbon-containing particles in water (e.g. Chughtai et al., 1996). The low soluble Fe concentrations, and hence low fractional Fe solubility, derived from the heavily caked soot filters between the 21 and 27 June could reflect the physical properties of

5 fresh black carbon, i.e., fresh black carbon does not disperse in water thus any soluble Fe associated with fresh black carbon is not dispersed in the water leach. However, combustion aerosols are known to have a high factional Fe solubility (Sholkovitz et al., 2012) and often these studies are based on aerosols collected shipboard in the remote open ocean where aerosol Fe has undergone atmospheric transport and aging. Therefore, the fractional Fe solubility in these studies is not directly comparable to fresh combustion Fe reported in this study. Combustion aerosols can become more soluble with transport (Ito, 2015). There

is a growing body of work that suggests aerosol Fe solubility can be enhanced by cloud chemistry and acid processing (Meskhidze et al., 2003;Spokes et al., 1994;Desboeufs et al., 1999;Kumar et al., 2010;Hoffer et al., 2005). Ito and Shi (2015) show that enhanced Fe solubility of dust could be related to reactive organic species such as oxalate, in cloud water, which contains Fe-binding functionalities such as humic-like substances from biomass burning. Fresh black carbon-containing particles are initially insoluble, but can be aged into a form that disperses in water following uptake of sulphuric acid and

secondary organic material via condensation and coagulation in the atmosphere. Black carbon particles can act as a seed for condensation and the components that condense on the surface of the black carbon particles are water soluble. Aged black carbon-containing particles can also disperse in water through the oxidation with functional groups during atmospheric transport over the remote ocean (Lohmann et al., 2000;Decesari et al., 2002;Chughtai et al., 1991;Chughtai et al., 1996). If the surface of black carbon particles becomes hydrophilic with the coating of water soluble species in the atmosphere, soluble Fe

bound to the black carbon-containing particles would disperse in water and be captured in the water soluble Fe leach. This could explain why, i) the fractional Fe solubility was higher in distally sourced fire event "E" compared to proximal fire events "FG" and "HI", and ii) the relatively high fractional Fe solubility observed in long-range transported aerosol to the remote Southern Ocean and Antarctica (Conway et al., 2015;Gaspari et al., 2006;Winton et al., 2015;Winton et al., 2016b). The impact of transport time and distance on fractional Fe solubility should be further investigated.

Oxalate modification of mineral dust could be a leading factor enhancing the fractional Fe solubility of mineral dust (Ito and Shi, 2015) in regions where there are high oxalate concentrations in the atmosphere, such as the tropics. The higher fractional Fe solubility during dust events compared to fire events at Gunn Point could be related to biomass burning derived-oxalate enhancing the solubility of Australian mineral dust that has been transported to the Northern Territory and mixed with biomass

burning plumes (Fig. 6). The fractional Fe solubility of mineral dust in the Southern Hemisphere is around 0.5-2 % (Sholkovitz et al., 2012), which is lower than our estimates of fractional Fe solubility during dust events in northern Australia. Only trace concentrations of oxalate are found in air masses in lower latitudes over Antarctica and the Southern Ocean (Keywood, 2007). Although, little is known about the enhancement of fractional Fe solubility in these pristine air masses (Chance et al., 2015;Winton et al., 2015), the concentrations of oxalate could well be too low to influence fractional iron solubility of mineral





dust. Therefore, the mean biomass burning enhanced mineral dust fractional Fe solubility of ~8 % represents an upper bound of mineral dust fractional Fe solubility in the Australian tropics during the dry season as illustrated in Fig. 7.

The relatively low fractional Fe solubility in mixed dust and fresh biomass burning aerosols at Gunn Point, compared to other
estimates of combustion aerosols (up to 60 %), could be related to the short aging time of fresh biomass burning in our samples. Ito (2015) model the transport of soluble Fe derived from combustion sources. The model indicates relatively low fractional Fe solubility near the sources of biomass burning and coal combustion. The fractional Fe solubility becomes higher as aerosols are transported to the open ocean. Therefore, transport time and distance could be an import factor in fractional Fe solubility. Alternatively, the relatively low fractional Fe solubility could be related to the type of biomass in northern Australia (i.e.
savannah that is comprised of eucalypt dominated woodlands (10-30 % foliage cover) and open-forests (30-70 % foliage cover), with a diverse woody sub-canopy and grassy ground cover (Edwards et al., 2015)). There is a wide range of fractional Fe solubility estimates for combustion aerosol in the literature and more studies are required to understand the fractional Fe solubility in different biomass types. On the other hand, low fractional Fe solubility during fire events and the inverse relationship between elemental carbon and soluble Fe questions whether biomass burning could be a potential bioavailable
source of Fe. Clearly future work should be directed towards the solubility of Fe in fresh and aged smoke plumes.

## 5. Conclusions

Co-emissions of mineral dust and aged biomass burning from Australia are potential sources of soluble Fe to the Southern Ocean and Australian tropical waters. During the SAFIRED campaign, northern Australia in the dry season 2014, there was considerable temporal variability in soluble and total PM10 aerosol Fe concentrations that reflect coincident mineral dust and
fresh smoke sources. Fractional Fe solubility was relatively high throughout the campaign and ranged from 2 to 12 %. During dust events, the fractional Fe solubility was greatest (12 %), however, decreased to 3 % during an extreme biomass burning event. Whilst mineral dust supplied soluble iron throughout the campaign, the lower fractional Fe solubility in the large biomass burning event suggests that the primary factor controlling soluble iron was the presence of mineral dust. Due to the hydrophobic nature of fresh black carbon, biomass burning derived soluble iron itself may not be a direct source of soluble
iron to the ocean. Nevertheless, biomass burning species can enhance the soluble iron chemistry in mixed aerosols. Iron in dust may be more soluble in the tropics, compared to higher latitudes, due to the higher concentrations of biomass burning derived reactive organic species in the atmosphere, such as oxalate, and their potential to enhance the fractional Fe solubility of mineral dust. Soluble Fe could be further enhanced during atmospheric transport and aging of black carbon-containing particles that could explain the relatively high episodic fractional Fe solubility observed in long-range transportable aerosol Fe
to the Southern Ocean and Antarctica. In addition, black carbon-containing particles can act as a surface for aerosol iron to bind to during transport. Biomass burning constitutes a large fraction of the aerosol loading over the tropics which has the potential to modulate soluble Fe and trigger nitrogen fixing toxic algal blooms. Such, toxic algal blooms have harmful





consequences for humans and other vertebrates. The understanding of the factors that initiate algal blooms needs to be improved (Law et al., 2011;Abram et al., 2003), especially over tropical regions where inputs of biomass burning to the ocean are predicted to increase over the next century (Keywood et al., 2013).

**Data availability**

The trace element dataset is available through the Curtin University Research Data repository http://doi.org/10.4225/06/5671012A48C2A.

**Author contribution**

V.H.L.W., R.E. and A.R.B. designed the research; V.H.L.W. and M.D. collected the aerosol filters; V.H.L.W., R.E. and A.R.B. prepared the trace element filter samples, analyzed and evaluated the data; P.W.S. prepared the elemental carbon and major

ion filter samples and analyzed the data; M.D. and C.P.W. collected and analyzed the carbon monoxide data; S.C. and A.W. collected and analyzed the radon data; M.M and M.K. analyzed the particle size distribution data; M.K., P.W.S., S.C., A.W., M.M., M.D., C.P.W. and V.H.L.W. contributed to the field campaign; V.H.L.W. prepared the manuscript with contributions from all co-authors.

**Acknowledgments**

This paper is a contribution to the Savannah Fires in the Early Dry Season (SAFIRED) campaign. This project was funded through Curtin University (RES-SE-DAP-AW-47679-1 to R.E), the University of Tasmania (B0019024 to A.R.B.), the Australian Research Council (FT130100037 to A.R.B.), the Antarctic Climate and Ecosystems (ARC CRC) and the CSIRO. Access to HR-ICP-MS instrumentation at Curtin University was facilitated through ARC LIEF funding (LE130100029). V.H.L.W. would like to acknowledge the following scholarship support: Australian Postgraduate Award, Curtin Research

Scholarship, CUPSA data collection grant and Curtin University Publication Scholarship. Thank you to CSIRO Ocean and Atmosphere for the use of their high-volume aerosol sampler and to the Bureau of Meteorology for the use of their site throughout the campaign. Thank you to the SAFIRED team, in particular to Rob Gillet and Jason Ward for assisting with the changeover of daily aerosol filters and to Brad Atkinson from the Bureau of Meteorology for assistance at Gunn Point. Thank you to Sylvester Werczynski from ANSTO for obtaining the back trajectories from HYSPLIT and cataloguing them in a

database for use with the Gunn Point data. Thank you to Pier van der Merwe for technical support with aerosol digestions. Wind roses were created using OpenAir package in R and meteorological data from the Bureau of Meteorology.



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



**Table 1:Trace metal aerosol samples (PM10) collected at Gunn Point, dry season 2014.**

| Sample time | [a]Start date/ time | [a]Finish date/ time | Total sampling duration (min) | [b]Total air volume (m$^3$) | [c]Total air volume (m$^3$) |
|---|---|---|---|---|---|
| GP1 | 4/06/2014 11:05 | 5/06/2014 10:45 | 1117 | 1102 | 1122 |
| GP2 | 5/06/2014 11:10 | 6/06/2014 12:30 | 1498 | 1458 | 1490 |
| GP3 | 6/06/2014 12:55 | 7/06/2014 11:30 | 1355 | 1531 | 1572 |
| GP4 | 7/06/2014 11:54 | 8/06/2014 8:47 | 1253 | 1416 | 1445 |
| GP5 | 8/06/2014 9:08 | 9/06/2014 10:21 | 1512 | 1708 | 1715 |
| GP6 | 9/06/2014 10:42 | [d]10/06/2014 | 711 | 802 | 806 |
| GP7 | 10/06/2014 2:41 | 11/06/2014 9:25 | 1112 | 1256 | 1278 |
| GP8 | 11/06/2014 10:16 | 12/06/2014 10:43 | 1467 | 1658 | 1669 |
| GP9 | 12/06/2014 11:03 | 13/06/2014 9:04 | 1322 | 1494 | 1518 |
| GP10 | 13/06/2014 10:02 | 14/06/2014 10:56 | 1494 | 1688 | 1729 |
| GP11 | 14/06/2014 11:31 | 15/06/2014 10:29 | 1378 | 1556 | 1584 |
| GP12 | 15/06/2014 11:11 | 16/06/2014 10:16 | 1385 | 1565 | 1572 |
| GP13 | 16/05/2014 10:45 | 17/06/2014 9:39 | 1374 | 1552 | 1550 |
| GP14 | 17/06/2014 10:04 | 18/06/2014 10:18 | 1454 | 1643 | 1635 |
| GP14 | 17/06/2014 10:04 | 18/06/2014 10:18 | 1454 | 1643 | 1635 |
| GP15 | 18/06/2014 11:16 | 19/06/2014 10:15 | 1379 | 1559 | 1577 |
| GP16 | 19/06/2014 10:41 | 20/06/2014 9:36 | 1375 | 1554 | 1584 |
| GP17 | 20/06/2014 9:59 | 21/06/2014 9:58 | 1439 | 1626 | 1647 |
| GP18 | 21/06/2014 10:31 | 22/06/2014 8:31 | 1319 | 1490 | 1508 |
| GP19 | 22/06/2014 8:55 | 23/06/2014 9:32 | 1476 | 1668 | 1669 |
| GP20 | 23/06/2014 10:39 | 24/06/2014 9:37 | [c]1269 | 1429 | 1441 |
| GP21 | 24/06/2014 10:04 | 25/06/2014 9:11 | 1387 | 1567 | 1569 |
| GP22 | 25/06/2014 9:42 | 26/06/2014 9:05 | 1403 | 1499 | 1497 |
| GP23 | 26/06/2014 9:46 | 27/06/2014 8:27 | 1360 | 1537 | 1536 |

[a]Australian Central Standard Time

[b]Total air volume calculated using STP.

[c]Total air volume corrected to ambient temperature and pressure.

5   [d]Fuse blown.



**Table 2: Aerosol iron concentrations and dry deposition fluxes for Gunn Point PM10 aerosols.**

| Sample | Soluble Fe concentration (ng m$^{-3}$) | ± | *Soluble Fe flux (µmol m$^{-2}$ d$^{-1}$) | ± | Total Fe concentration (ng m$^{-3}$) | ± | *Total Fe flux (µmol m$^{-2}$ d$^{-1}$) | ± |
|---|---|---|---|---|---|---|---|---|
| GP1 | 30 | 2 | 0.9 | 0.5 | 286 | 9 | 9 | 4 |
| GP2 | 37 | 1 | 1.1 | 0.6 | 384 | 12 | 12 | 6 |
| GP3 | 49 | 2 | 1.5 | 0.8 | 598 | 19 | 18 | 9 |
| GP4 | 78 | 3 | 2.4 | 1.2 | 1164 | 37 | 36 | 18 |
| GP5 | 140 | 6 | 4.3 | 3.2 | 1158 | 37 | 36 | 18 |
| GP6 | 66 | 3 | 2.0 | 1.0 | 823 | 26 | 25 | 13 |
| GP7 | 20 | 0.8 | 0.6 | 0.3 | 236 | 7 | 7 | 4 |
| GP8 | 35 | 1 | 1.1 | 0.5 | 374 | 12 | 12 | 6 |
| GP9 | 35 | 1 | 1.1 | 0.5 | 385 | 12 | 12 | 6 |
| GP10 | 46 | 2 | 1.4 | 0.7 | 539 | 17 | 17 | 8 |
| GP11 | 92 | 4 | 2.8 | 1.4 | 928 | 29 | 29 | 14 |
| GP12 | 52 | 2 | 1.6 | 0.8 | 677 | 21 | 21 | 10 |
| GP13 | 31 | 1 | 1.0 | 0.5 | 331 | 10 | 10 | 5 |
| GP14 | 23 | 0.9 | 0.7 | 0.3 | 192 | 6 | 6 | 3 |
| GP15 | 15 | 0.6 | 0.5 | 0.2 | 123 | 4 | 4 | 2 |
| GP16 | 7 | 0.3 | 0.2 | 0.1 | 60 | 2 | 2 | 1 |
| GP17 | 67 | 3 | 2.1 | 1.0 | 1129 | 36 | 35 | 17 |
| GP18 | 13 | 0.5 | 0.4 | 0.2 | 247 | 8 | 8 | 4 |
| GP19 | 7 | 0.3 | 0.2 | 0.1 | 231 | 7 | 7 | 4 |
| GP20 | 13 | 0.5 | 0.4 | 0.2 | 301 | 10 | 9 | 5 |
| GP21 | 7 | 0.3 | 0.2 | 0.1 | 225 | 7 | 7 | 3 |
| GP22 | 10 | 0.4 | 0.3 | 0.1 | 399 | 13 | 12 | 6 |
| GP23 | 11 | 0.4 | 0.3 | 0.1 | 314 | 10 | 10 | 5 |

*The uncertainty in dry deposition fluxes was calculation by propagation of error of the analytical uncertainty and uncertainty in the deposition velocity assumed to be 50 %



**Table 3: Enrichment factor analysis of Gunn Point aerosol samples.**

|  | Sr | Pb | Al | V | Cr | Mn | Fe | K |
|---|---|---|---|---|---|---|---|---|
| GP1 | 0.37 | 2.57 | 0.60 | 1.81 | 1.22 | 0.53 | 1.12 | 1.72 |
| GP2 | 0.37 | 3.18 | 0.76 | 2.24 | 1.54 | 0.67 | 1.28 | 1.91 |
| GP3 | 0.40 | 2.32 | 0.68 | 2.09 | 1.41 | 0.88 | 1.13 | 1.23 |
| GP4 | 0.34 | 1.38 | 0.64 | 1.93 | 1.26 | 0.88 | 1.07 | 0.67 |
| GP5 | 0.24 | 1.83 | 0.55 | 1.77 | 1.28 | 0.68 | 0.97 | 0.64 |
| GP6 | 0.31 | 3.42 | 0.59 | 1.72 | 1.34 | 1.03 | 1.03 | 2.17 |
| GP7 | 0.71 | 8.81 | 0.59 | 1.81 | 1.88 | 1.44 | 1.02 | 5.50 |
| GP8 | 0.42 | 3.77 | 0.66 | 2.08 | 1.34 | 0.72 | 1.15 | 1.49 |
| GP9 | 0.29 | 4.25 | 0.58 | 1.85 | 1.38 | 0.60 | 1.08 | 1.63 |
| GP10 | 0.23 | 3.42 | 0.67 | 2.29 | 7.93 | 0.49 | 1.16 | 1.37 |
| GP11 | 0.17 | 1.22 | 0.73 | 2.00 | 1.32 | 0.44 | 1.14 | 0.34 |
| GP12 | 0.43 | 2.05 | 0.64 | 1.95 | 1.48 | 1.47 | 1.14 | 1.23 |
| GP13 | 0.39 | 5.32 | 0.62 | 1.57 | 1.09 | 1.22 | 1.07 | 1.07 |
| GP14 | 0.68 | 6.55 | 0.64 | 1.67 | 1.16 | 1.48 | 1.08 | 2.30 |
| GP15 | 0.76 | 6.60 | 0.65 | 2.84 | 3.37 | 1.07 | 0.99 | 1.79 |
| GP16 | 1.43 | 2.39 | 0.62 | 1.77 | 0.99 | 1.40 | 0.88 | 1.49 |
| GP17 | 0.33 | 0.79 | 0.72 | 1.73 | 4.22 | 0.23 | 0.91 | 0.25 |
| GP18 | 0.52 | 1.63 | 0.67 | 2.09 | 1.34 | 0.52 | 1.04 | 0.48 |
| GP19 | 0.66 | 12.46 | 0.63 | 1.78 | 1.55 | 1.22 | 1.00 | 2.65 |
| GP20 | 1.38 | 8.05 | 0.68 | 1.60 | 1.25 | 4.70 | 1.10 | 4.79 |
| GP21 | 0.50 | 4.65 | 0.69 | 1.90 | 1.32 | 1.43 | 1.16 | 2.63 |
| GP22 | 1.16 | 5.72 | 0.71 | 1.96 | 1.37 | 2.18 | 1.12 | 3.92 |
| GP23 | 1.04 | 4.25 | 0.69 | 1.92 | 1.22 | 1.77 | 1.04 | 1.76 |



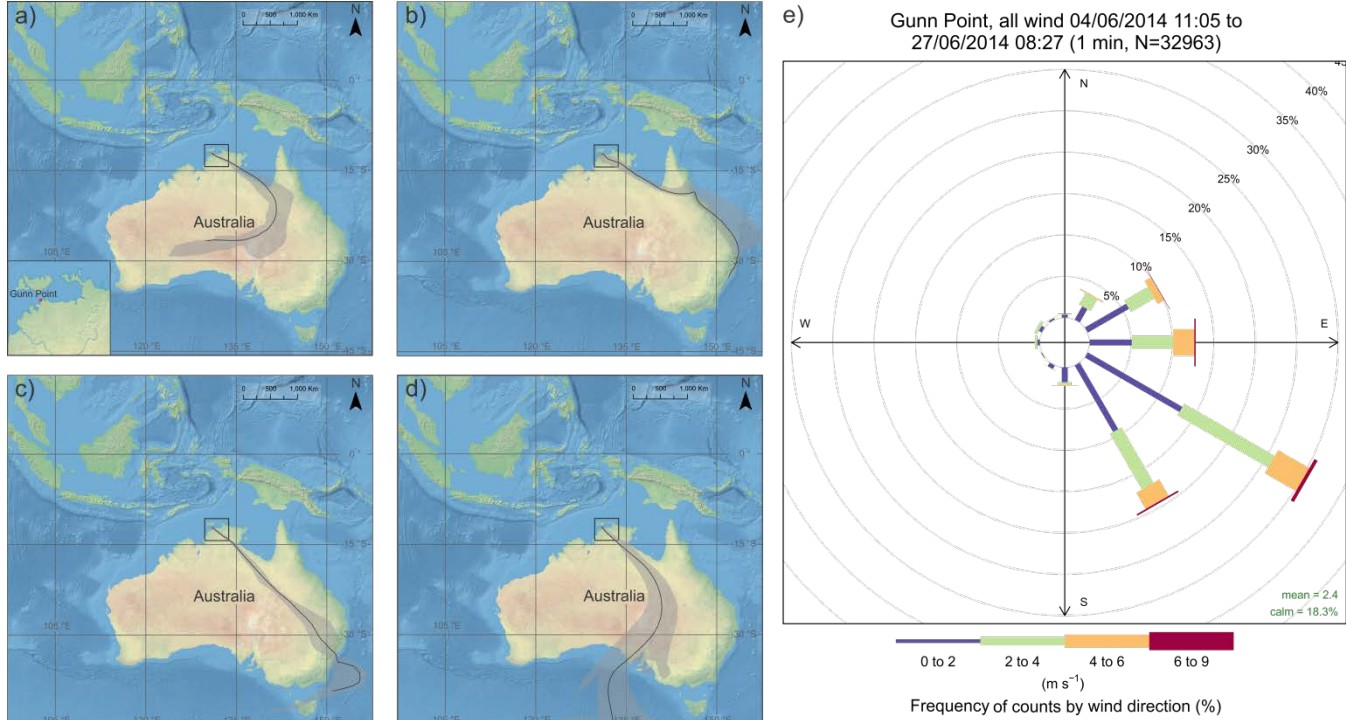

**Figure 1: Location of Gunn Point study site. Insert of the Northern Territory showing the Gunn Point site and the predominant air mass fetch regions for the June 2014 Gunn Point campaign. a) Inland, low population density. b) Coastal, moderate urban/industrial activity. c) Major urban/industrial activity. d) Southerly. d) Wind rose corresponding to the aerosol sampling duration. Wind rose created using the OpenAir package in R (Carslaw and Ropkins, 2012;Carslaw, 2014).**





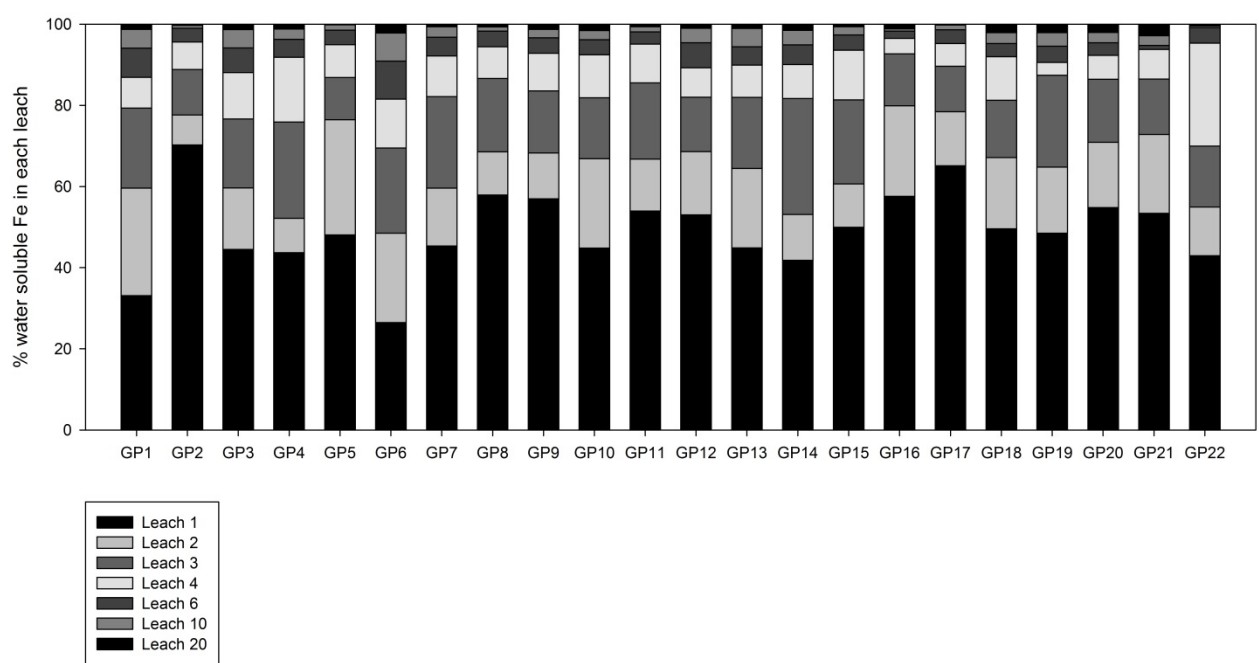

Figure 2: Fraction of water soluble iron in sequential leaches of all PM10 samples from the SAFIRED campaign. Details of samples GP1 to GP23 can be found in Table 1.



**Figure 3: Time series of a) diurnal radon, b) advective radon, and total PM10 trace element concentrations c) Ti, d) Al, e) Fe, f) V, and g) Na.**





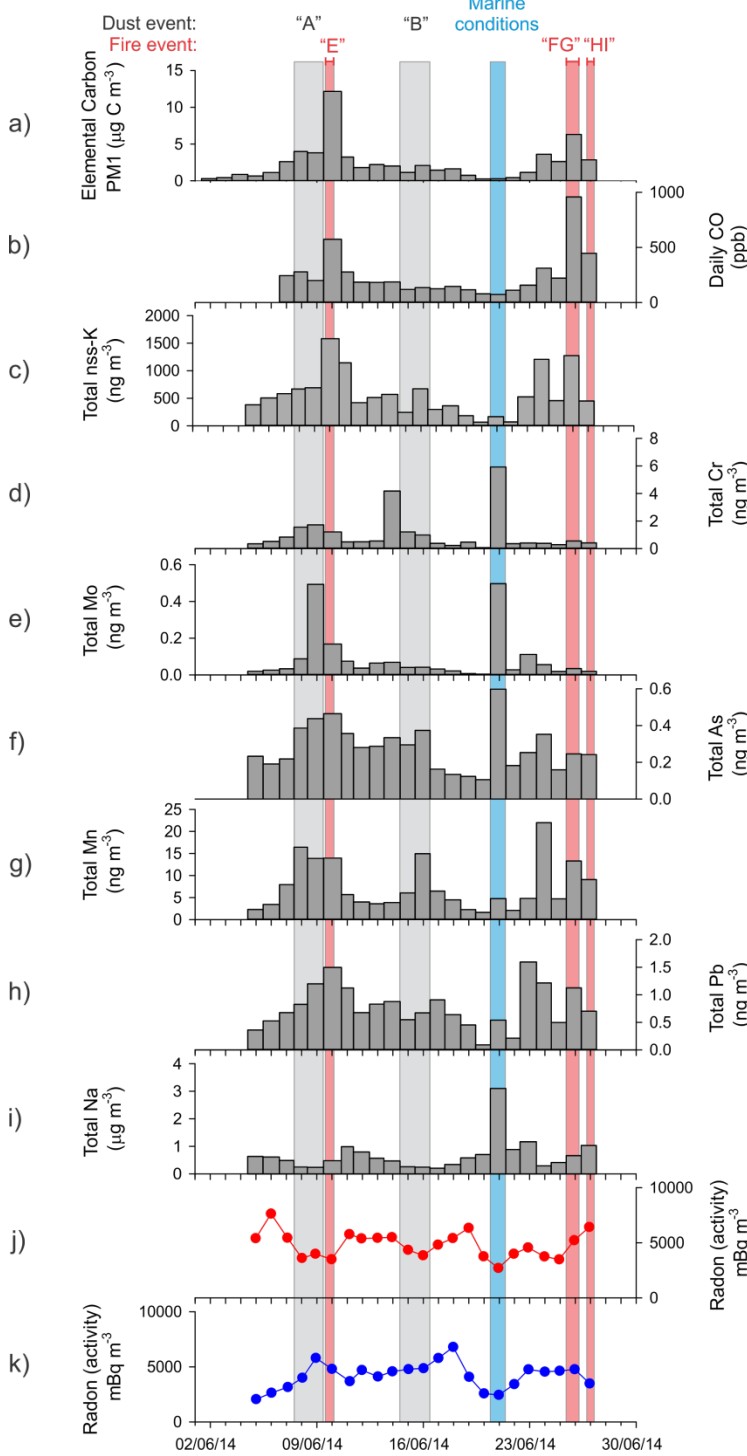

**Figure 4: Time series of a) elemental carbon, b) carbon monoxide, and total PM10 trace element concentrations c) nss-K, d) Cr, e) Mo, f), As, g) Mn, h) Pb, i) Na, and j) diurnal radon, k) advective radon.**







**Figure 5: Time series of a) elemental carbon, b) carbon monoxide, c) total nss-K, soluble trace element concentrations, d) Al, e) Ti, f), Fe, g) Mn, h) V and i) diurnal radon, j) advective radon.**





**Figure 6: Time series of fractional Fe solubility, oxalate, elemental carbon and levoglucosan concentrations.**







**Figure 7: Gunn Point total PM10 aerosol iron mass concentration versus fractional Fe solubility superimposed on the Southern Hemispheric aerosol iron data set (Sholkovitz et al., 2012; and references within;Bowie et al., 2009;Gao et al., 2013;Winton et al., 2015). Boxes highlight the two clusters. Cluster 1: moderate fractional Fe solubility and moderate total aerosol iron loading between the 4 and 19 June, and represents the upper bound of dust fractional Fe solubility for the topical dry season. Cluster 2: low fractional Fe solubility and low total aerosol iron loading between the 24 and 26 June.**