# Peer review of "Dry season aerosol iron solubility in tropical northern Australia"

_Atmospheric Chemistry and Physics, 2016_

## Referee Comment (RC1) · Anonymous Referee #1 · 2 Jul 2016

The manuscript reports results from analyzing gaseous and particulate matter collected from a field campaign in Australia during the dry season. The objective is to correlate the fraction of soluble iron to markers from natural and anthropogenic sources of aerosols, and to compare the amount of soluble iron in aerosols from this site to other sites reported in the literature.

Overall the manuscript is organized and describes the trends in the figures and tables in detail. The major weakness of the manuscript is that there are key observations mentioned in the manuscript from data presented in a manuscript in preparation by Mallet et al. (See pages 4, 8, 12, 13, and 14). Hence, either the authors show the relevant data from that manuscript to support their arguments or wait on submitting this manuscript till the one by Mallet et al. is published.

The comments below are minor and intended to improve the readability of the manuscript:

- The abstract is too long and contains a reference, which is not recommended. Revise for brevity and to highlight the main findings of the manuscript, - Show correlation plots for soluble iron concentration versus oxalate and versus black carbon to support the discussion section, - On page 3, combine paragraphs 2 and 3, - On page 7, line 15, list the major cations and anions, - Revise the manuscript for long sentences (more than 3 lines in length) and shorten them, - On page 9, switch the order of sentences between lines 5 and 10 to present data in Fig. 3 then Fig. 5, - Revise the manuscript for consistency in citing figures as 'Figure' or 'Fig.', - On page 10, comparisons with previous related work is summarized. It would be better to make a graph that highlights the similarities among different sites, - On page 15, cite a recent review article published on the surface and bulk chemistry of iron that is related to the discussion in this section: "Al-Abadleh, H. A., A Review on the Bulk and Surface Chemistry of Iron in Atmospherically-relevant Systems Containing Humic Like Substances. RSC Adv. 2015, 5, 45785 - 45811.", - Use different types of marker shapes for Figure 7, - Rearrange figures in Figure S1 to enlarge font size in numbers and axis labels, - Use different line styles for Figure S2,

---

## Referee Comment (RC2) · Anonymous Referee #2 · 14 Jul 2016

This manuscript reports the results of a study of aerosol and gas composition at Gunn Point, Northern Territories, Australia, during a period in the dry season that was periodically impacted by biomass burning events. A large amount of good quality data appears to have been generated and the subject of the manuscript – the impact of biomass burning on aerosol trace metal solubility – is one that is relevant to the journal. However, the manuscript suffers from a number of problems and will require major revisions before it is suitable for publication. Currently the text is marred by inaccuracy, inconsistency, vagueness and lacks focus. It does not do justice to a worthwhile and interesting study. I only list my major concerns here, but I would hope that the authors would submit a much more self-consistent and concise revised manuscript.

Accuracy: "During dust event "A" fractional Fe solubility peaked at 12%" (page 10, line 23 and repeated on page 12 and 16). According to Figure 6, the upper limit of fractional

Fe solubility during event "A" was ~8%. On the figure the 12% value corresponds to burning event "E". Which is correct, the figure or the text?

"This same set of total trace elements (Cr, As, Mo, and V) also peaked on the 9 June during fire event "E" (Fig. 5)." This statement also does not stand up to scrutiny. Of the elements listed, only As has a (slightly) higher concentration in the sample associated with event "E" than in the previous sample. The data referred to is presented in Figures 3 & 4, not 5.

I would urge the authors to be much more cautious in their use and interpretation of enrichment factors. There are many pitfalls associated with their use (including the choice of reference material and reference element, and whether the reference material is actually representative of mineral dust in the atmosphere – see Reimann and De Caritat (2000) and Arimoto et al. (1989) for brief descriptions of some of these pitfalls). In light of these problems, most authors choose to use much higher values of EF to describe individual samples as "enriched", e.g. >5-10 (e.g. Duce et al., 1983; Gelado-Caballero et al., 2012). The values of EF presented in Table 3 are almost all rather low (there is only one value above 10) and therefore the case for describing most elements as enriched is weak. The author's attempts to link these low values to potential anthropogenic emission sources several thousand kilometres distant from their sampling site (e.g. in Brisbane) are untenable (even without considering the uncertainties in the air mass back trajectories used to make the link).

The value for dry deposition velocity of Fe used here has been taken from Duce et al. (1991) and Baker et al., (2003) – see page 8. These authors estimated the uncertainty in dry deposition velocities to be a factor of 2-3. Here the uncertainty in this parameter is estimated to be +/- 50% (footnote to Table 2). How do the authors justify this?

"biomass burning contributes around 60-70 % of the total soluble Fe deposition, where the dust contribution is less (40-50 %)" (page 14). On the face of it, these numbers don't add up.

Consistency: Interpretation of the manuscript would be easier if the air mass types encountered during the campaign were identified in a consistent manner. I was extremely puzzled by the apparent contradiction of the statement "No trajectories fell in the 'mainly oceanic' category" and the labelling on Figures 3-6 of a period of "Marine conditions". Only on page 13 does it become clear that "marine conditions" actually corresponds to type b (coastal, moderate urban/industrial activity), as defined on page 7. Understanding of the influence of air mass source would also be improved if the periods influenced by each identified type were marked on those figures.

It would be helpful if the way that details of the many measurement and analysis techniques presented here could be harmonised as much as possible, especially for the two types of aerosol collection. Were the quartz filters really handled in a trace metal clean laboratory?

On page 11 trajectory type c is described as including "major cities and industrial areas in Brisbane". On page 12 the same trajectory type is described as "times when the air mass passed over central Australian desert and low population areas of inland Australia".

Vagueness: Section 3.4 is good example of this (but there are several others). This manuscript is based on a rich multi-parameter dataset. Surely the authors can find more scientific descriptions than "x was high at the same time as y".

"annual dry season" (page 4 & 14). What does this mean?

Focus: The title of the manuscript indicates a study focussed on aerosol iron solubility. I think some of the discussion of peripheral parameters could be removed without any risk of degrading the manuscript. As an example, the potential source of As from volcanoes in these samples (page 13) is of very little relevance. On the other hand, it is quite possible that the fractional solubility data for the other trace metals measured during the campaign could prove to be very informative. There is a little discussion of this for Al, Ti and Mn, but the actual data are not shown. Apparently the trends in

fractional solubility variation over the campaign are not unique to Fe. Does this imply that the factors that regulate Fe solubility in these samples also regulate the solubility of Al and Ti? What about the other trace elements? Please add a plot of fractional solubility over time (similar to Figures 3 – 6) for all the elements possible, even if this has to be put in the Supplement.

There are some lines of discussion that I feel have been left incomplete, especially in relation to the influence of biomass burning on the results obtained. Apparently (page 8) the nss-K/Fe ratio during fire events was higher than at other times. Since nss-K is a well-established marker for biomass burning, does that result imply that biomass burning is an insignificant source of Fe to the aerosol? The answer would depend on the emission ratios for K and Fe from biomass burning. There are numerous laboratory-based studies of biomass burning emissions. Please could K/Fe ratios for such studies be added as a comparison. In a similar vein, on page 11 the authors state that enrichment factors for Fe were between 0.9 and 1.3 and that therefore "anthropogenic pollution was not a dominant source of Fe to Gunn Point". What implications does this result have for biomass burning as a source of Fe? Perhaps it implies that biomass burning is also relatively unimportant, but perhaps not. Entrainment of fine soil particles into flames and smoke plumes could add Fe to the aerosol without affecting enrichment factors. Again, emission factors from laboratory studies could be informative here, at least with regard to direct emission from combusted biomass. Although the study of Guieu et al. (2005) on biomass burning-related emission of Fe in the Mediterranean is cited several times, I would like to see the authors make a direct comparison between the results of the two studies. Are they consistent with one another?

Authorship: There is absolutely no need for the authors to refer to themselves through the use of Personal Communications. All of these instances should be removed.

Citations: Please update the citations of Fomba et al, ACPD, 2012 and Ito & Shi, ACPD, 2016 to their final published forms in ACP (2013 and 2016 respectively).

Methodology: Filtration of water soluble trace metal leachates. The authors only state that one sample (GP22) was filtered through 0.2 um filters after leaching (page 5) and that leachates (the non-0.2 um filtered leachates or also the filtered leachate of GP22?) contained visible particles after settling (page 14). If leachates contained particles, what efforts were made to disperse these within the solution immediately prior to analysis by ICP-MS?

Were quartz filters washed before use? If so, how?

Please give a little more detail on the method used to determine total metal concentrations, e.g. which acids were used for digestion.

References Arimoto, R., Duce, R. A., and Ray, B. J.: Concentrations, sources and air-sea exchange of trace elements in the atmosphere over the Pacific Ocean, in: Chemical Oceanography. SEAREX: The sea/air exchange program, edited by: Riley, J. P., Chester, R., and Duce, R. A., Academic Press, London, 107-149, 1989.

Duce, R. A., Arimoto, R., Ray, B. J., Unni, C. K., and Harder, P. J.: Atmospheric trace elements at Enewetak Atoll: 1, Concentrations, sources and temporal variability, J. Geophys. Res., 88, 5321-5342, 1983.

Gelado-Caballero, M. D., López-García, P., Prieto, S., Patey, M. D., Collado, C., and Hérnández-Brito, J. J.: Long-term aerosol measurements in Gran Canaria, Canary Islands: Particle concentration, sources and elemental composition, J. Geophys. Res., 117, D03304, 10.1029/2011JD016646, 2012.

Reimann, C., and De Caritat, P.: Intrinsic flaws of element enrichment factors (EFs) in environmental geochemistry, Environmental Science & Technology, 34, 5084-5091, 10.1021/es001339o, 2000.

---

## Referee Comment (RC3) · Anonymous Referee #3 · 19 Jul 2016

**Review of "Dry season aerosol iron solubility in tropical northern Australia", by V. H. L. Winton, R. Edwards, A. R. Bowie, M. Keywood, A. G. Williams, S. Chambers, P. W. Selleck, M. Desservettaz, M. Mallet, and C. Paton-Walsh (ACP-2016-419).**

**General Comments**

This study presents results from a measurement campaign in northern Australia (SAFIRED), and in particular focuses on the solubility of iron in aerosol sampled from air masses influenced by dust and biomass burning events, both local and distant.

The factors influencing aerosol iron solubility are still not fully understood, and this work provides interesting and valuable insights into the solubility of iron associated with biomass burning aerosol. In particular, the authors provide evidence that iron in freshly emitted combustion aerosol is not particularly soluble, which in turn suggests that atmospheric aging itself plays a significant role in observations of soluble iron in plumes sampled far from combustion sources.

Unfortunately, the paper in its current form is difficult to follow, and I find its conclusions overstated as written. Since my concerns relate to the interpretation and presentation of the data rather than the data itself, however, I do recommend publication in *Atmospheric Chemistry and Physics* after substantial revisions addressing the comments below.

In the following sections, I outline my specific scientific questions/issues regarding the manuscript, organized by manuscript section. Although I've attempted to collate my technical/presentation issues in the final section, which addresses the overall clarity of the manuscript, this wasn't always possible, since I sometimes found cosmetic and substantive issues difficult to disentangle.

**Methodology**

**P5L11**   "following the ultra-clean methodology" … is this a reference to a specific methodology?

**Section 2.3.1**   From what I understand, there are a number of published methods for determining iron solubility, and a number of operational definitions of iron solubility. How does the strategy presented here fit within the literature/precedent on this topic? Specifically, in P6L1, what does "Leaches 5–etc. *were estimated*" mean? Finally, why was GP22 in particular chosen for filtration?

**P7L17**   Is this the correct baking temperature?

**Results**

**P8L13**   How is nss-K (*i.e.* versus total K) determined here?  Is the assumption that *all* K is nss-K, since no trajectories were "mainly oceanic" (P8L1)?

**P8L22**   The nss-K/Fe ratio during non-fire events (1.3) is higher than that quoted for both crustal aerosol and polluted aerosol.  What is the overall significance of this comparison?

**P8L29**   This discussion could benefit from some reference to the literature.

**P9L1**   Would you expect the same behaviour for samples collected under different conditions (*i.e.* during non-fire events)?  And, what is the overall significance of this observation?

**Section 3.4**   These results could be presented more clearly—the correlations described aren't actually shown in the figures, and the figures are discussed out of order.  Also, Na seems to be repeated in Figures 3 and 4.  As a minor note, the section seems to be mis-titled, as it also includes discussion of elements other than iron. More importantly, I'm confused by the reference to "marine conditions" here, as earlier in the paper it was stated that no trajectories were mainly oceanic.

**Discussion**

**P9L29**   I'm wondering about the first line of Section 4.1—technically, isn't the *non-fire* PM10 Fe concentration also similar to the Brazilian fire aerosol studies, given that there isn't an obvious correlation between PM10 Fe and fire event presence/absence?  Is this a useful comparison? Also, according to Figure 3, the maximum PM10 Fe concentration actually appears to be ~0.8 ug/m3?  In addition, what is the overall significance of the comparisons to other fire events if the results aren't scaled to total PM10 / some proxy for proximity to the actual fires?

**Section 4.3**   I find this discussion confusing as written and very difficult to follow. In particular, I'm confused by this statement: "These estimates are similar to dust estimates around 0.5-2 % at relatively high Fe mass concentrations." Perhaps it might make sense to merge this section with the discussion of the fire/dust events … why is fractional Fe solubility highest during the dust event?

**P12L11**   This paragraph is vague as written.  "Lower enrichment factors"—lower than what? In addition, I wonder if the EF data should be presented in the results section.  There is a lot of information presented here (*e.g.* detailed discussion of Pb, V, Mn sources) and in my opinion, it dilutes the overall message of the paper.

**Section 4.5.1**   Why is the fractional Fe solubility so high during the dust event?  This is discussed in Section 4.6 to some extent, but perhaps that discussion should be moved here?

**Section 4.5.2**   Again, I don't disagree with anything presented here, but I do think that discussion of Cr/Mo/As is distracting—more time is spent discussing sources of these elements than in discussing the fractional solubility of Fe in this air mass, which seems skewed.

**Section 4.5.3**   The second paragraph discusses relationships between fractional Fe solubility and various biomass burning tracers, and states that fractional Fe solubility was low during fire events FG and HI (local), but higher during fire event E (distant).  However, the fractional Fe solubility in the days prior to fire events FG and HI was also low—the fire plume doesn't appear to be the only source of low-solubility Fe.  More generally, to me, it doesn't appear from the data in Figure 6 that the fractional Fe solubility during fire events is significantly different from that on the surrounding days.  An explanation of this would be useful, I think.

**Section 4.6**   The first paragraph of this section is more suited for the introduction. In P14L17, can the conclusions for GP22 be extrapolated to the rest of the campaign, when biomass burning aerosol concentrations were low and the aerosol population was primarily dust-influenced? In P15L1, again, it seems to me that the fractional Fe solubility was generally lower over the last week of the campaign, rather than that "the fractional Fe solubility was lower when fresh EC was high". Perhaps this discussion would be better framed along the lines of "Despite the influx of aerosol from

proximal fires, the fractional Fe solubility remained low"? In P16L13–14, you state an "inverse relationship between elemental carbon and soluble Fe", yet in P13L25, you state that there is no relationship between these measurements.  This is directly contradictory.

**Conclusions**

Generally, I think that the conclusions are somewhat overstated and not particularly well-supported by the data, *e.g.*:

- P16L21 states that the fractional Fe solubility decreased to 3% *during* the biomass burning event, whereas the data shows that the solubility was also low prior to the event.
- The discussion of the influence of oxalate on fractional Fe solubility would make sense in the introduction, but I don't think that this data set as presented supports this conclusion (are the oxalate concentrations high enough to support this conclusion?  P15L26, etc., don't argue this convincingly).
- P16L28 is, in my mind, overstated—the fractional Fe solubility in fire event E is not substantially elevated over background conditions (*e.g.* the first sample in Figure 6).

**Manuscript clarity**

I often found the manuscript difficult to follow. I provide examples here, primarily from the introduction, but suggest that the manuscript be carefully edited/reorganized to maximize flow/clarity/organization on both the paragraph- and section-level scales.

- The first paragraph of the introduction states first that iron availability limits nitrogen fixation in nutrient-poor tropical waters (P2L12) and later that nitrogen fixation is iron-limited in 75% of the world oceans (P2L22)—these thoughts could be combined.
- The second paragraph of the introduction presents examples of aerosol-induced toxic algal blooms as if it were a new topic, with no link made to P2L11.
- The third paragraph of the introduction discusses two Australian aeolian dust paths, but in insufficient detail to picture them—what do "northwest" and "southeast" mean, exactly?
- The paragraph on black carbon terminology seems misplaced (P3L21–29) — perhaps this can be included in the methods section?
- The sentence beginning on the last line of P3 is unclear as written.
- In P4L3, what is meant by "smoke"?  This seems vague.
- The final paragraph of the introduction is unconvincing to me (as written—I don't mean to imply that the study isn't worthwhile!) ... as written, it seems as though a goal is to better understand dust inputs to the Southern Ocean, whereas the rest of the introduction (and the abstract) has focused on the tropical waters north of Australia
- Table 1 could perhaps be moved to the supplemental information (or, just the most important information from it could be included in Table 2)
- Tenses are sometimes inconsistent (*e.g.* Section 2.4 has both present and past tense).
- References to personal communications in Section 3.2 should be removed.
- A number of typos throughout, *e.g.* "particulate patter" (P8L17), "biomass burring" (P13L24)
- The use of A/B for dust events and A/B/C/D for trajectory types is very confusing, and requires a lot of flipping back and forth between text and figures.
- In P11L13, "contamination" doesn't seem an appropriate word.

---

## Author Comment (AC1)

**Manuscript: ACP-2016-419**

**Author response to referee #1**

We thank the reviewer for their time and constructive comments for improving this manuscript. We have reproduced the reviewer comments below and have appended our responses to each of their queries in italics. Technical revisions and minor changes were highly appreciated and were followed as suggested. We have only reproduced the additional comments in the referee's supplement if rephrasing of entire sentences and additional data was requested or in a very few cases where we disagreed with the suggestions. Page numbers refer to the revised version of the manuscript.

**Anonymous Referee #1 interactive comment**

The manuscript reports results from analyzing gaseous and particulate matter collected from a field campaign in Australia during the dry season. The objective is to correlate the fraction of soluble iron to markers from natural and anthropogenic sources of aerosols, and to compare the amount of soluble iron in aerosols from this site to other sites reported in the literature. Overall the manuscript is organized and describes the trends in the figures and tables in detail. The major weakness of the manuscript is that there are key observations mentioned in the manuscript from data presented in a manuscript in preparation by Mallet et al. (See pages 4, 8, 12, 13, and 14). Hence, either the authors show the relevant data from that manuscript to support their arguments or wait on submitting this manuscript till the one by Mallet et al. is published.

*Comment: Snice the original submission of this manuscript, Mallet et al. has been submitted to ACP Discussions (acp-2016-731). Several companion papers from the Savannah Fires in the Early Dry Season (SAFIRED) 2014 campaign in northern Australia have also been submitted or published in ACP Discussions, including a study of biomass burning aerosol aging (Milic et al., 2016) and savannah fire emission factors (acp-2016-708).*

*M. Desservettaz, C. Paton-Walsh, D. W. T. Griffith, G. Kettlewell, M. D. Keywood, M. V. Vanderschoot, J. Ward, M. D. Mallet, A. Milic, B. Milievic, Z. D. Ristovski, D. Howard, G. C. Edwards, and B. Atkinson. Emission factors of trace gases and particles in tropical northern Australia. MS No.: acp-2016-708.*

*M. Mallet, M. Desservettaz, B. Miljevic, A. Milic, Z. Ristovski, J. Alroe, L. Cravigan, R. Jayaratne, C. Paton-Walsh, D. Griffith, S. Wilson, G. Kettlewell, M. van der Schoot, P. Selleck, F. Reisen, S. Lawson, J. Ward, J. Harnwell, M. Cheng, R. Gillett, S. Molloy, D. Howard, P. Nelson, A. Morrison, G. Edwards, A. Williams, S. Chambers, S. Werczynski, L. Williams, H. Winton, B. Atkinson, X. Wang, and M. Keywood. An overview of the Savannah Fires in the Early Dry Season (SAFIRED) 2014 campaign in northern Australia. MS No.: acp-2016-731.*

*Milic, A., Mallet, M. D., Cravigan, L. T., Alroe, J., Ristovski, Z. D., Selleck, P., Lawson, S. J., Ward, J., Desservettaz, M. J., Paton-Walsh, C., Williams, L. R., Keywood, M. D., and Miljevic, B.: Aging of aerosols emitted from biomass burning in northern Australia, Atmos. Chem. Phys. Discuss., 2016, 1-24, 10.5194/acp-2016-730, 2016.*

The comments below are minor and intended to improve the readability of the manuscript:

- The abstract is too long and contains a reference, which is not recommended. Revise for brevity and to highlight the main findings of the manuscript

*Comment: The abstract has been shortened to highlight the main findings and the reference removed.*

- Show correlation plots for soluble iron concentration versus oxalate and versus black carbon to support the discussion section

*Comment: A correlation plot for fractional Fe solubility versus elemental carbon can be found in panel a) of Fig. S1. We have added panel b) to Fig. S1 of fractional Fe solubility versus oxalate.*

- On page 3, combine paragraphs 2 and 3

*Comment: Paragraphs 2 and 3 have been combined.*

- On page 7, line 15, list the major cations and anions

*Comment: As this paper uses oxalate and levoglucosan measurements. The heading has been changed to "2.6. Oxalate and levoglucosan"*

- Revise the manuscript for long sentences (more than 3 lines in length) and shorten them

*Comment: Long sentences have been edited throughout the manuscript.*

- On page 9, switch the order of sentences between lines 5 and 10 to present data in Fig. 3 then Fig. 5

*Comment: Sentences have been rearranged so that the references to figures are in chronological order.*

- Revise the manuscript for consistency in citing figures as 'Figure' or 'Fig.'

*Comment: We have followed the ACP guidelines for authors regarding the abbreviation of figures, i.e., "Fig." is used when it appears in running text and "Figure" is used at the beginning of a sentence.*

- On page 10, comparisons with previous related work is summarized. It would be better to make a graph that highlights the similarities among different sites

*Comment: Our fractional Fe solubility and total aerosol Fe data are compared to other Southern Hemispheric aerosol iron studies in Fig. 7. Different regions are colour coded to make visual comparison clearer.*

- On page 15, cite a recent review article published on the surface and bulk chemistry of iron that is related to the discussion in this section: "Al-Abadleh, H. A., A Review on the Bulk and Surface Chemistry of Iron in Atmospherically-relevant Systems Containing Humic Like Substances. RSC Adv. 2015, 5, 45785 - 45811.",

*Comment: This article has been cited on page 15 lines, 13-15. Thank you for this suggestion.*

*"In a review of Fe chemistry in systems containing humic-like substances, Al-Abadleh (2015) highlights the complexation and dissolution processes in which aerosol Fe solubility is enhanced with organic compounds."*

- Use different types of marker shapes for Figure 7

*Comment: Different shapes have been applied to the data points in Fig. 7.*

- Rearrange figures in Figure S1 to enlarge font size in numbers and axis labels

*Comment: The font has been enlarged in Fig. S1.*

- Use different line styles for Figure S2

*Comment: Different line styles have been applied to Fig. S2.*

**References**

Milic, A., Mallet, M. D., Cravigan, L. T., Alroe, J., Ristovski, Z. D., Selleck, P., Lawson, S. J., Ward, J., Desservettaz, M. J., Paton-Walsh, C., Williams, L. R., Keywood, M. D., and Miljevic, B.: Aging of aerosols emitted from biomass burning in northern Australia, Atmos. Chem. Phys. Discuss., 2016, 1-24, 10.5194/acp-2016-730, 2016.

---

## Author Comment (AC2)

**Manuscript: ACP-2016-419**

**Author response to referee #2**

We thank the reviewer for their time and constructive comments for improving this manuscript. We have reproduced the reviewer comments below and have appended our responses to each of their queries in italics. Technical revisions and minor changes were highly appreciated and were followed as suggested. We have only reproduced the additional comments in the referee's supplement if rephrasing of entire sentences and additional data was requested or in a very few cases where we disagreed with the suggestions. Page numbers refer to the revised version of the manuscript.

**Anonymous Referee #2 interactive comment**

Accuracy: "During dust event "A" fractional Fe solubility peaked at 12%" (page 10, line 23 and repeated on page 12 and 16). According to Figure 6, the upper limit of fractional Fe solubility during event "A" was ~8%. On the figure the 12% value corresponds to burning event "E". Which is correct, the figure or the text?

*Comment: The text is correct but the fractional Fe solubility data in Fig. 6 was wrongly plotted. The fractional Fe solubility in dust event "A" is 12 %. Fig. 6 has been replotted with the correct fractional Fe solubility data. The data in all plots have consequently been checked.*

"This same set of total trace elements (Cr, As, Mo, and V) also peaked on the 9 June during fire event "E" (Fig. 5)." This statement also does not stand up to scrutiny. Of the elements listed, only As has a (slightly) higher concentration in the sample associated with event "E" than in the previous sample. The data referred to is presented in Figures 3 & 4, not 5.

*Comment: Total As and Pb have higher concentrations during fire event "E" (Fig. 4). The text has been re-written on page 9, line 26-28.*

I would urge the authors to be much more cautious in their use and interpretation of enrichment factors. There are many pitfalls associated with their use (including the choice of reference material and reference element, and whether the reference material is actually representative of mineral dust in the atmosphere – see Reimann and De Caritat (2000) and Arimoto et al. (1989) for brief descriptions of some of these pitfalls). In light of these problems, most authors choose to use much higher values of EF to describe individual samples as "enriched", e.g. >5-10 (e.g. Duce et al., 1983; GeladoCaballero et al., 2012). The values of EF presented in Table 3 are almost all rather low (there is only one value above 10) and therefore the case for describing most elements as enriched is weak. The author's attempts to link these low values to potential anthropogenic emission sources several thousand kilometres distant from their sampling site (e.g. in Brisbane) are untenable (even without considering the uncertainties in the air mass back trajectories used to make the link).

*Comment: We agree with referee #2 and note the limitations of using enrichment factors (EF) in page 11, lines 25-27 and include the references suggested (Arimoto et al., 1989;Reimann and Caritat, 2000). We follow the Duce et al. (1983);Gelado-Caballero et al. (2012) classification of EF, i.e., "enriched EFs" between 5 and 10 and "low EF" less than 5. Following this, we have modified our interpretation of the EFs in page 11, lines 27-30. This*

*suggests that the origin of the material is similar to the upper continental crust for all elements except Pb (GP19) and Cr (GP9) which have EFs greater than 5. As the new classification of EFs discounts potential anthropogenic emission sources, we have deleted the discussion concerning enriched elements of Cr, Mn, Pb, K, As, Mo and V.*

The value for dry deposition velocity of Fe used here has been taken from Duce et al. (1991) and Baker et al., (2003) – see page 8. These authors estimated the uncertainty in dry deposition velocities to be a factor of 2-3. Here the uncertainty in this parameter is estimated to be +/- 50% (footnote to Table 2). How do the authors justify this?

*Comment: To be consistent, we have used the uncertainty of dry deposition velocities (a factor of 2-3) reported in Duce et al. (1991);Baker et al. (2003). This had been updated on page 10, lines 18-19 and in Table 2.*

"biomass burning contributes around 60-70 % of the total soluble Fe deposition, where the dust contribution is less (40-50 %)" (page 14). On the face of it, these numbers don't add up.

*Comment: The sentence has been re-written on page 3, line 14-16.*

*"Ito (2015) shows that biomass burning derived soluble Fe contributes substantial inputs of soluble Fe to tropical and Southern Ocean waters downwind of Australia."*

Consistency: Interpretation of the manuscript would be easier if the air mass types encountered during the campaign were identified in a consistent manner. I was extremely puzzled by the apparent contradiction of the statement "No trajectories fell in the 'mainly oceanic' category" and the labelling on Figures 3-6 of a period of "Marine conditions". Only on page 13 does it become clear that "marine conditions" actually corresponds to type b (coastal, moderate urban/industrial activity), as defined on page 7. Understanding of the influence of air mass source would also be improved if the periods influenced by each identified type were marked on those figures.

*Comment: Air mass types are now referred to consistently throughout the manuscript. The reference to "mainly oceanic" trajectories has been deleted (page 8, line 4) as it is irrelevant, i.e., no trajectories were solely over the ocean during the campaign.*

*"marine conditions" are now referred to as "coastal conditions."*

*"Marine conditions" in Figs. 3-6 have been renamed "coastal conditions" to be consistent with the air mass trajectory during that period - type b) "coastal, moderate urban/industrial activity" (Fig. 1b).*

*The heading "4.5.2. Anthropogenic and sea spray sources in marine conditions" has been renamed "4.5.2. Sea spray sources during coastal conditions".*

*In Figs. 3-6 we have added the trajectory types to the labels highlighting the periods of dust events, biomass burning events and coastal conditions.*

It would be helpful if the way that details of the many measurement and analysis techniques presented here could be harmonised as much as possible, especially for the two types of aerosol collection. Were the quartz filters really handled in a trace metal clean laboratory?

*Comment: We have rearranged the methodology by putting information for each type of filter (quartz and Whatman 41) under the relevant headings. For the blank filters, we have stated which type of blank is associated with which type of measurement (page 4, lines 25-32). The quartz filters were analysed at CSIRO Ocean and Atmosphere and this is now clear. The section about sample preparation for trace metal work has been moved to the heading "2.3. Trace element analysis" (page 5, lines 2-5).*

On page 11 trajectory type c is described as including "major cities and industrial areas in Brisbane". On page 12 the same trajectory type is described as "times when the air mass passed over central Australian desert and low population areas of inland Australia".

*Comment: For consistency, we have changed the name for trajectory type "c" to "inland, major urban/industrial activity." This has been updated in the Fig. 1 caption and throughout the manuscript. Dust events are associated with trajectories that passed over central Australian desert (page 12, lines 6-8). "and low population areas of inland Australia" has been deleted from the sentence as the trajectories also passed over major urban/industrial regions in Sydney and Adelaide.*

Vagueness: Section 3.4 is good example of this (but there are several others). This manuscript is based on a rich multi-parameter dataset. Surely the authors can find more scientific descriptions than "x was high at the same time as y".

*Comment: Section 3.4 has been rewritten.*

"annual dry season" (page 4 & 14). What does this mean?

*Comment: In northern Australia, the dry season ranges from May/June (early dry season) to October-November (late dry season) (Andersen et al., 2005). We have deleted "annual" and defined the dry season on page 3, lines 22-23.*

Focus: The title of the manuscript indicates a study focussed on aerosol iron solubility. I think some of the discussion of peripheral parameters could be removed without any risk of degrading the manuscript. As an example, the potential source of As from volcanoes in these samples (page 13) is of very little relevance. On the other hand, it is quite possible that the fractional solubility data for the other trace metals measured during the campaign could prove to be very informative. There is a little discussion of this for Al, Ti and Mn, but the actual data are not shown. Apparently the trends in fractional solubility variation over the campaign are not unique to Fe. Does this imply that the factors that regulate Fe solubility in these samples also regulate the solubility of Al and Ti? What about the other trace elements? Please add a plot of fractional solubility over time (similar to Figures 3 – 6) for all the elements possible, even if this has to be put in the Supplement.

*Comment: We agree with referee #2 and have removed discussion around peripheral anthropogenic parameters, such as As, Mo and Cr on page 11 from lines 27. "Anthropogenic" has been removed from the heading 4.5.2 which now focusses on coastal conditions. The heading has been renamed to "4.5.2. Sea spray sources during coastal conditions."*

*Regarding the fractional Fe solubility of other trace metals, we have added plots of fractional solubility of Al, Ti and Mn in Fig. 6. The temporal variability of fractional Al and Ti is similar to Fe, i.e., fractional solubility of Al, Ti and Fe were highest between 4 and 20 June and suddenly dropped after 20 June during the largest and most proximal fires (as discussed on*

*page 11, lines 4-11). On the other hand, Mn is relatively soluble and appears unrelated to the fractional solubility of Fe, Al, Ti. As far as we are aware no data for these elements has been collected at biomass burning source regions during the dry season. Thus, we cannot speculate that the factors that regulate Fe solubility in these samples may also regulate the fractional solubility of Al and Ti but not Mn. We have eluded to this on page 16, lines 11-16 and suggest future work should investigate these elements in conjunction.*

There are some lines of discussion that I feel have been left incomplete, especially in relation to the influence of biomass burning on the results obtained. Apparently (page 8) the nss-K/Fe ratio during fire events was higher than at other times. Since nss-K is a well-established marker for biomass burning, does that result imply that biomass burning is an insignificant source of Fe to the aerosol? The answer would depend on the emission ratios for K and Fe from biomass burning. There are numerous laboratory based studies of biomass burning emissions. Please could K/Fe ratios for such studies be added as a comparison. In a similar vein, on page 11 the authors state that enrichment factors for Fe were between 0.9 and 1.3 and that therefore "anthropogenic pollution was not a dominant source of Fe to Gunn Point". What implications does this result have for biomass burning as a source of Fe? Perhaps it implies that biomass burning is also relatively unimportant, but perhaps not. Entrainment of fine soil particles into flames and smoke plumes could add Fe to the aerosol without affecting enrichment factors. Again, emission factors from laboratory studies could be informative here, at least with regard to direct emission from combusted biomass. Although the study of Guieu et al. (2005) on biomass burning-related emission of Fe in the Mediterranean is cited several times, I would like to see the authors make a direct comparison between the results of the two studies. Are they consistent with one another?

*Comment: More studies are required to determine the influence of biomass burning on soluble iron both at the source and in long range air masses downwind of biomass burning sources. In terms of the influence of biomass burning in this study, we have tried to focus the discussion, conclusion and abstract around the following:*

*During the large burning events in this study, fresh elemental carbon seems to be an insignificant source of Fe to the aerosol (it has a low Fe solubility, it is initially hydrophobic and contains relatively little Fe). We have already pointed this out on page 13, lines 1-2 "During these fire events, total PM10 Fe and soluble Fe concentrations and fractional Fe solubility (~3 %) were considerably less than the average concentration and fractional solubility throughout the campaign." However, biomass burning may be important through indirect processes such as oxalate modification, as a substrate to transport long range aerosol Fe, and through the entrainment of fine soil into the plume. The nss-K/Fe ratio and the EFs for Fe in the plumes suggest that elemental carbon is relatively unimportant. Biomass burning could be important indirectly via soil entrainment into the smoke plume. We have already discussed the process of how Fe can be incorporated into biomass burning plumes on page 13, line 29-32. The values of the nss-K/Fe ratios and the EF could be explained by the process. We have added this to the discussion on page 11, lines 31-32 and page 13, lines 21-24.*

*There are many papers that give emission factors for K (Gaudichet et al., 1995;Akagi et al., 2011) however, we are not aware of any that give emission factors of Fe from fires. Iron is not included in Akagi 's review.*

*Nss-K/Fe ratios for other studies have been added on page 8, lines 24-26 for comparison to our results (Paris et al., 2010;Srinivas et al., 2012).*

*We have made a comparison to the Guieu et al. (2005) study on biomass burning-related emission of Fe in the Mediterranean on page 16, lines 6-9. While the studies are not directly comparable, both suggest that biomass burning is not a major source of soluble iron to the ocean.*

Authorship: There is absolutely no need for the authors to refer to themselves through the use of Personal Communications. All of these instances should be removed.

*Comment: References of personal communications have been removed and updated with published or submitted papers.*

Citations: Please update the citations of Fomba et al, ACPD, 2012 and Ito & Shi, ACPD, 2016 to their final published forms in ACP (2013 and 2016 respectively).

*Comment: Citations have been updated.*

Methodology: Filtration of water soluble trace metal leachates. The authors only state that one sample (GP22) was filtered through 0.2 um filters after leaching (page 5) and that leachates (the non-0.2 um filtered leachates or also the filtered leachate of GP22?) contained visible particles after settling (page 14). If leachates contained particles, what efforts were made to disperse these within the solution immediately prior to analysis by ICP-MS?

*Comment: The instantaneous soluble iron fraction from these fine particles was leached into solution during the flow through leach. To avoid sucking fine particles into the ICP-MS, the leachates were left to settle in centrifuge tubes overnight before analysis. For total iron measurements, concentrated $HNO_3$ and HF were used to digest aerosol particles into solution.*

Were quartz filters washed before use? If so, how?

*Comment: The quartz filters were not washed. Pall tissue quartz filters are extremely low in soluble ion and carbohydrate blank concentrations. The oxalate blank, calculated using average collection volume of 1575 $m^3$, is 0.0009 $\mu g\ m^{-3}$ with a MDL of 0.0002. Levoglucosan has no detectable blank with a MDL of 0.00002 $\mu g\ m^{-3}$. These blank concentrations have been added in page 7, lines 12-14.*

Please give a little more detail on the method used to determine total metal concentrations, e.g. which acids were used for digestion.

*Comment: All information for filter digestions can be found in Winton et al. (2016). We have added the acids (HNO3 and HF) used in the digestion on page 5, line 30.*

**References**

Akagi, S., Yokelson, R. J., Wiedinmyer, C., Alvarado, M., Reid, J., Karl, T., Crounse, J., and Wennberg, P.: Emission factors for open and domestic biomass burning for use in atmospheric models, Atmospheric Chemistry and Physics, 11, 4039-4072, 2011.
Andersen, A. N., Cook, G. D., Corbett, L. K., Douglas, M. M., Eager, R. W., RUSSELL-SMITH, J., Setterfield, S. A., Williams, R. J., and Woinarski, J. C.: Fire frequency and biodiversity conservation in Australian tropical savannas: implications from the Kapalga fire experiment, Austral Ecology, 30, 155-167, 2005.

Arimoto, R., Duce, R. A., and Ray, B. J.: Concentrations, sources and air-sea exchange of trace elements in the atmosphere over the Pacific Ocean, Chemical Oceanography, 10, 107-149, 1989.

Baker, A., Kelly, S., Biswas, K., Witt, M., and Jickells, T.: Atmospheric deposition of nutrients to the Atlantic Ocean, Geophysical Research Letters, 30, 2003.

Duce, R., Arimoto, R., Ray, B., Unni, C., and Harder, P.: Atmospheric trace elements at Enewetak Atoll: 1. Concentrations, sources, and temporal variability, Journal of Geophysical Research: Oceans, 88, 5321-5342, 1983.

Duce, R. A., Liss, P. S., Merrill, J. T., Atlas, E. L., Buat-Menard, P., Hicks, B. B., Miller, J. M., Prospero, J. M., Arimoto, R., Church, T. M., Ellis, W., Galloway, J. N., Hansen, L., Jickells, T. D., Knap, A. H., Reinhardt, K. H., Schneider, B., Soudine, A., Tokos, J. J., Tsunogai, S., Wollast, R., and Zhou, M.: The atmospheric input of trace species to the world ocean, Global Biogeochem. Cycles, 5, 193-259, 10.1029/91gb01778, 1991.

Gaudichet, A., Echalar, F., Chatenet, B., Quisefit, J., Malingre, G., Cachier, H., Buat-Menard, P., Artaxo, P., and Maenhaut, W.: Trace elements in tropical African savanna biomass burning aerosols, Journal of Atmospheric Chemistry, 22, 19-39, 1995.

Gelado-Caballero, M. D., López-García, P., Prieto, S., Patey, M. D., Collado, C., and Hérnández-Brito, J. J.: Long-term aerosol measurements in Gran Canaria, Canary Islands: Particle concentration, sources and elemental composition, Journal of Geophysical Research: Atmospheres, 117, 2012.

Paris, R., Desboeufs, K., Formenti, P., Nava, S., and Chou, C.: Chemical characterisation of iron in dust and biomass burning aerosols during AMMA-SOP0/DABEX: implication for iron solubility, Atmospheric Chemistry and Physics, 10, 4273-4282, 2010.

Reimann, C., and Caritat, P. d.: Intrinsic flaws of element enrichment factors (EFs) in environmental geochemistry, Environmental Science & Technology, 34, 5084-5091, 2000.

Srinivas, B., Sarin, M., and Kumar, A.: Impact of anthropogenic sources on aerosol iron solubility over the Bay of Bengal and the Arabian Sea, Biogeochemistry, 110, 257-268, 2012.

Winton, H., Bowie, A., Keywood, M., van der Merwe, P., and Edwards, R.: Suitability of high-volume aerosol samplers for ultra-trace aerosol iron measurements in pristine air masses: blanks, recoveries and bugs, Atmos. Meas. Tech. Discuss., 2016, 1-32, 10.5194/amt-2016-12, 2016.

---

## Author Comment (AC3)

**Manuscript: ACP-2016-419**

**Author response to referee #3**

We thank the reviewer for their time and constructive comments for improving this manuscript. We have reproduced the reviewer comments below and have appended our responses to each of their queries in italics. Technical revisions and minor changes were highly appreciated and were followed as suggested. We have only reproduced the additional comments in the referee's supplement if rephrasing of entire sentences and additional data was requested or in a very few cases where we disagreed with the suggestions. Page numbers refer to the revised version of the manuscript.

**Anonymous Referee #3 interactive comment**

Methodology

P5L11 "following the ultra-clean methodology" … is this a reference to a specific methodology?

*Comment: We had added a reference for trace metal clean practices on page 5, lines 3-4:*

*"following trace metal clean practices (e.g. Cutter et al., 2010)."*

Section 2.3.1 From what I understand, there are a number of published methods for determining iron solubility, and a number of operational definitions of iron solubility. How does the strategy presented here fit within the literature/precedent on this topic? Specifically, in P6L1, what does "Leaches 5–etc. were estimated" mean? Finally, why was GP22 in particular chosen for filtration?

*Comment: Yes, there are a vast number of methods employed in the literature for estimating soluble iron concentrations of aerosols. These range from water soluble leaching methods (e.g. Buck et al., 2006), weak acid leaching methods (e.g. Shi et al., 2011;Baker et al., 2006), seawater leaching methods (e.g. Aguilar-Islas et al., 2010) to iron speciation methods (e.g. Spolaor et al., 2012;Chen and Siefert, 2004). The leaching methods vary in the leaching solution, volume and time (instantaneous to months). This variability gives rise to a number of operational definitions of soluble iron, and makes comparisons of results from different studies difficult. Although, there are few estimates of soluble iron in Australian dust and aerosols, for consistency, we use the instantaneous water leaching scheme that has previously been used for other studies of Australian aerosols (Winton et al., 2015;Winton et al., 2016). We have added the justification for using the instantaneous water soluble method on page 5, lines 7-11.*

*Due to the time and cost of analysing 20 leachates per sample, we only collected and analysed leaches 1-4, 6, 10 and 20. It has been widely reported in the literature that soluble iron concentrations exponentially decrease with subsequent leaches (e.g. Aguilar-Islas et al., 2010). We therefore, estimated the concentrations of the unanalysed leachates by fitting a power law curve to the soluble iron concentrations of leaches 1-4, 6, 10 and 20. "Leaches 5–etc. were estimated" has been rewritten as "Concentrations of leachates." We have clarified this on page 5, lines 11-18 and 20-23.*

*GP22 was chosen because it was particularly high in of soluble iron. This has been added on page 5, line 24.*

P7L17 Is this the correct baking temperature?

*Comment: The quartz filters were baked at 600°C and this has been corrected on page 7, line 8.*

Results

P8L13 How is nss-K (i.e. versus total K) determined here? Is the assumption that all K is nss-K, since no trajectories were "mainly oceanic" (P8L1)?

*Comment: The nss-K fraction of total K was obtained by subtracting the contribution of sea salt derived K from the measured K concentrations:*

*nss-K = K -0.037 x Na*

*where Na and K are the measured concentrations in aerosol samples and 0.037 is the K/Na ratio in sea-salt (Keene et al., 1986).*

*We have added this on page 6, lines 7-10.*

P8L22 The nss-K/Fe ratio during non-fire events (1.3) is higher than that quoted for both crustal aerosol and polluted aerosol. What is the overall significance of this comparison?

*Comment: The higher ratio for non-fire events suggests that even during the non-biomass burning events, biomass burning was still a contributing source of aerosol Fe. We have added this significance on page 8, lines 30-31.*

P8L29 This discussion could benefit from some reference to the literature.

*Comment: The exponential decrease of soluble Fe in sequential leaches is discussed on page 5, lines 11-12 and page, 9 lines 4-8 with reference to other studies who also observed this trend (e.g. Aguilar-Islas et al., 2010;Wu et al., 2007;Fishwick et al., 2014).*

P9L1 Would you expect the same behaviour for samples collected under different conditions (i.e. during non-fire events)? And, what is the overall significance of this observation?

*Comment: Yes, the proportion of soluble iron in <0.2 μm pool is likely to change between seasons as different sources (with different particle size distributions) turn on and off. For example, biomass burning has a finer particle size distribution to mineral dust. This has been discussed in the discussion concerning particle sizes and bio-availability on page 14, lines 17-19.*

Section 3.4 These results could be presented more clearly—the correlations described aren't actually shown in the figures, and the figures are discussed out of order. Also, Na seems to be repeated in Figures 3 and 4. As a minor note, the section seems to be mis-titled, as it also includes discussion of elements other than iron. More importantly, I'm confused by the reference to "marine conditions" here, as earlier in the paper it was stated that no trajectories were mainly oceanic.

*Comment: Section 3.4 has been rewritten. The figures are now discussed in order. Scatterplots showing the correlations have been added in the supplement (Fig. S3). Section 3.4 has been*

retitled *"Trace element mass concentrations"*. Na was repeated in Fig. 4 so the reader could see he high concentrations of total As, Cr and Mo during the marine conditions. We have deleted the Na plot from Fig. 4. *"marine conditions"* are now referred to as *"coastal conditions"* to be consistent with the air mass trajectory during that period - type b) *"coastal, moderate urban/industrial activity"* (Fig. 1b).

Discussion

P9L29 I'm wondering about the first line of Section 4.1—technically, isn't the non-fire PM10 Fe concentration also similar to the Brazilian fire aerosol studies, given that there isn't an obvious correlation between PM10 Fe and fire event presence/absence? Is this a useful comparison? Also, according to Figure 3, the maximum PM10 Fe concentration actually appears to be ~0.8 ug/m3? In addition, what is the overall significance of the comparisons to other fire events if the results aren't scaled to total PM10 / some proxy for proximity to the actual fires?

*Comment: Yes, the total Fe concentrations throughout the campaign are similar to the Brazilian fires. We have replaced "during fire events" with "during the campaign" on page 13, line 6. The maximum total Fe concentration throughout the campaign was 1.2 $\mu g \ m^{-3}$ and the highest total Fe concentration for fire events was 0.82 $\mu g \ m^{-3}$. The Alta Floresta, Amazon Forest, Brazil is at a similar latitude to the Gunn Point site. As far as we know, there are no other estimates of total Fe concentrations in biomass burning aerosols in Australia, thus we have compared our results to other studies from tropical regions that experience seasonal biomass burning (e.g. the Alta Floresta) and the open ocean. In tropical regions, it is interesting that the concentrations of total Fe are similar at the source (Gunn Point and Alta Floresta) but decrease over the open ocean (Atlantic). We have rearranged the section 4.1 to make this clearer.*

Section 4.3 I find this discussion confusing as written and very difficult to follow. In particular, I'm confused by this statement: "These estimates are similar to dust estimates around 0.5-2 % at relatively high Fe mass concentrations." Perhaps it might make sense to merge this section with the discussion of the fire/dust events … why is fractional Fe solubility highest during the dust event?

*Comment: Further details have been added to this section. The sentence concerning dust estimates has been deleted. We have left the section on fractional Fe solubility where it is and refer to these values in the discussion sections of dust and biomass burning events. We discuss factors that can enhance the Fe solubility in biomass burning derived particles in section 4.6 (particle size, atmospheric processing, hydrophobic black carbon, modification with acidic species, aerosol aging). The factors that regulate fractional Fe solubility in these samples may also regulate the solubility of Al and Ti and this is discussed on page 16, lines 11-16.*

P12L11 This paragraph is vague as written. "Lower enrichment factors"—lower than what? In addition, I wonder if the EF data should be presented in the results section. There is a lot of information presented here (e.g. detailed discussion of Pb, V, Mn sources) and in my opinion, it dilutes the overall message of the paper.

*Comment: The section on EF has be focussed on Fe and peripheral elements (Pb,V, Mn) have been removed. Please see response to reviewer's 2 comment on this topic.*

Section 4.5.1 Why is the fractional Fe solubility so high during the dust event? This is discussed in Section 4.6 to some extent, but perhaps that discussion should be moved here?

*Comment: Factors that influence fractional Fe solubility are discussed in Section 4.6 and we refer the reader to that section (on page 12, line 12).*

Section 4.5.2 Again, I don't disagree with anything presented here, but I do think that discussion of Cr/Mo/As is distracting—more time is spent discussing sources of these elements than in discussing the fractional solubility of Fe in this air mass, which seems skewed.

*Comment: We have focused the manuscript around fractional Fe solubility, and removed much of the discussion around anthropogenic elements (Cr Mo and As) as they have little influence on fractional Fe solubility throughout the campaign. Please refer to our response to the comment made by reviewer 2 on this topic.*

Section 4.5.3 The second paragraph discusses relationships between fractional Fe solubility and various biomass burning tracers, and states that fractional Fe solubility was low during fire events FG and HI (local), but higher during fire event E (distant). However, the fractional Fe solubility in the days prior to fire events FG and HI was also low—the fire plume doesn't appear to be the only source of low-solubility Fe. More generally, to me, it doesn't appear from the data in Figure 6 that the fractional Fe solubility during fire events is significantly different from that on the surrounding days. An explanation of this would be useful, I think.

*Comment: Yes, the fractional Fe solubility is fairly constant ~3 % during the 21-27 June (page 13, line 9). During this time the filters were caked in soot and oxalate, levoglucosan and nss-K concentrations were high. Although we have marked on the figures the most intense fire events as determined from CO data, biomass burning occurs constantly throughout this region at this time. The campaign was heavily influenced by thousands of wild and prescribed bushfires (Milic et al., 2016). In the second paragraph, we focus on the period of low fractional Fe solubility (21-27 June) rather than fire events "FG" and "HI." This is explained on page 13, line 10.*

Section 4.6 The first paragraph of this section is more suited for the introduction. In P14L17, can the conclusions for GP22 be extrapolated to the rest of the campaign, when biomass burning aerosol concentrations were low and the aerosol population was primarily dust-influenced? In P15L1, again, it seems to me that the fractional Fe solubility was generally lower over the last week of the campaign, rather than that "the fractional Fe solubility was lower when fresh EC was high". Perhaps this discussion would be better framed along the lines of "Despite the influx of aerosol from proximal fires, the fractional Fe solubility remained low"? In P16L13–14, you state an "inverse relationship between elemental carbon and soluble Fe", yet in P13L25, you state that there is no relationship between these measurements. This is directly contradictory.

*Comment: The first paragraph has been moved to the introduction.*

*A discussion of fine particles observed in sample GP22 has been expanded on page 14, line 17-19. The sample GP22 occurred at the time of high biomass burning aerosols, and thus this sample is more indicative of particles during low fractional Fe solubility.*

*We have replaced the opening sentence on page 15, line 1 with that suggested by the reviewer.*

*On page 15, line 32, we have deleted "inverse relationship."*

Conclusions

Generally, I think that the conclusions are somewhat overstated and not particularly well supported by the data, e.g.:

• P16L21 states that the fractional Fe solubility decreased to 3% during the biomass burning event, whereas the data shows that the solubility was also low prior to the event.

*Comment: "event" replaced with "period."*

• The discussion of the influence of oxalate on fractional Fe solubility would make sense in the introduction, but I don't think that this data set as presented supports this conclusion (are the oxalate concentrations high enough to support this conclusion? P15L26, etc., don't argue this convincingly).

*Comment: Oxalate modulation has been shown to enhance fractional Fe solubility in other studies (Ito, 2015;Ito and Shi, 2016;Shi et al., 2011). We include this mechanism in our discussion as a factor that can influence mixed dust and biomass burning aerosol during transport. We do not see this enhancement in our data set as the aerosols are fresh but suggest it should be investigated in smoke plumes downwind of biomass burning sources.*

• P16L28 is, in my mind, overstated—the fractional Fe solubility in fire event E is not substantially elevated over background conditions (e.g. the first sample in Figure 6).

*Comment: This sentence refers to processes that could occur during transport and aging. It does not apply to what we are observing at the source.*

Manuscript clarity

I often found the manuscript difficult to follow. I provide examples here, primarily from the introduction, but suggest that the manuscript be carefully edited/reorganized to maximize flow/clarity/organization on both the paragraph- and section-level scales.

*Comment: The abstract and discussion has been edited and reorganised, along with sections in the methods and results.*

• The first paragraph of the introduction states first that iron availability limits nitrogen fixation in nutrient-poor tropical waters (P2L12) and later that nitrogen fixation is iron limited in 75% of the world oceans (P2L22)—these thoughts could be combined.

*Comment: These thoughts have been combined.*

• The second paragraph of the introduction presents examples of aerosol-induced toxic algal blooms as if it were a new topic, with no link made to P2L11.

*Comment: We have added a linking sentence on page 2, lines 14-16.*

*"Dissolved Fe can also be supplied to the surface ocean by vertical mixing, hydrothermal inputs and resuspension of marine sediments, in addition to wet and dry deposition from atmospheric sources (e.g. Tagliabue et al., 2010; Elrod et al., 2004; Mills et al., 2004)."*

• The third paragraph of the introduction discusses two Australian aeolian dust paths, but in insufficient detail to picture them—what do "northwest" and "southeast" mean, exactly?

*Comment: These dust paths travel northwest and southeast of the Australian continent into the adjacent waters. We have reworded the "northwest dust path" to the "transport of dust northwest of Australia."*

• The paragraph on black carbon terminology seems misplaced (P3L21–29) — perhaps this can be included in the methods section?

*Comment: The terminology of black carbon has been moved to Section 2.7 in the methods.*

• The sentence beginning on the last line of P3 is unclear as written.

*Comment: The sentence has been rewritten on page 4, line 11-12:*

*"During the sampling campaign, the strongest winds were predominantly from the southeast (Fig. 1)."*

• In P4L3, what is meant by "smoke"? This seems vague.

*Comment: "smoke" has been replaced with "biomass burning aerosols."*

• The final paragraph of the introduction is unconvincing to me (as written—I don't mean to imply that the study isn't worthwhile!) … as written, it seems as though a goal is to better understand dust inputs to the Southern Ocean, whereas the rest of the introduction (and the abstract) has focused on the tropical waters north of Australia

*Comment: We have removed reference to the study in Antarctica and focussed on tropical waters north of Australia rather than the Southern Ocean.*

• Table 1 could perhaps be moved to the supplemental information (or, just the most important information from it could be included in Table 2)

*Comment: Table 1 has been moved to Table S1 in the Supplement.*

• Tenses are sometimes inconsistent (e.g. Section 2.4 has both present and past tense).

*Comment: The manuscript has been checked for consistent tense.*

• References to personal communications in Section 3.2 should be removed.

*Comment: All references to personal communication of co-authors have been removed.*

• A number of typos throughout, e.g. "particulate patter" (P8L17), "biomass burring" (P13L24)

*Comment: Typos have been corrected.*

• The use of A/B for dust events and A/B/C/D for trajectory types is very confusing, and requires a lot of flipping back and forth between text and figures.

*Comment: In Figs. 3-6 we have added the trajectory types to the labels highlighting the periods of dust events, biomass burning events and coastal conditions.*

• In P11L13, "contamination" doesn't seem an appropriate word.

*Comment: The sentence has been rewritten:*

*"The EFs of Gunn Point filters are used to aid our interpretation of mineral dust versus other aerosol sources."*

**References**

Aguilar-Islas, A. M., Wu, J., Rember, R., Johansen, A. M., and Shank, L. M.: Dissolution of aerosol-derived iron in seawater: Leach solution chemistry, aerosol type, and colloidal iron fraction, Marine Chemistry, 120, 25-33, 2010.

Baker, A. R., Jickells, T. D., Witt, M., and Linge, K. L.: Trends in the solubility of iron, aluminium, manganese and phosphorus in aerosol collected over the Atlantic Ocean, Marine Chemistry, 98, 43-58, 2006.

Buck, C. S., Landing, W. M., Resing, J. A., and Lebon, G. T.: Aerosol iron and aluminum solubility in the northwest Pacific Ocean: Results from the 2002 IOC cruise, Geochem. Geophys. Geosyst., 7, Q04M07, 10.1029/2005gc000977, 2006.

Chen, Y., and Siefert, R. L.: Seasonal and spatial distributions and dry deposition fluxes of atmospheric total and labile iron over the tropical and subtropical North Atlantic Ocean, J. Geophys. Res., 109, D09305, 10.1029/2003jd003958, 2004.

Fishwick, M. P., Sedwick, P. N., Lohan, M. C., Worsfold, P. J., Buck, K. N., Church, T. M., and Ussher, S. J.: The impact of changing surface ocean conditions on the dissolution of aerosol iron, Global Biogeochemical Cycles, 2014GB004921, 10.1002/2014GB004921, 2014.

Ito, A.: Atmospheric Processing of Combustion Aerosols as a Source of Bioavailable Iron, Environmental Science & Technology Letters, 2, 70-75, 10.1021/acs.estlett.5b00007, 2015.

Ito, A., and Shi, Z.: Delivery of anthropogenic bioavailable iron from mineral dust and combustion aerosols to the ocean, Atmospheric Chemistry and Physics, 16, 85-99, 2016.

Keene, W. C., Pszenny, A. A., Galloway, J. N., and Hawley, M. E.: Sea-salt corrections and interpretation of constituent ratios in marine precipitation, Journal of Geophysical Research: Atmospheres, 91, 6647-6658, 1986.

Milic, A., Mallet, M. D., Cravigan, L. T., Alroe, J., Ristovski, Z. D., Selleck, P., Lawson, S. J., Ward, J., Desservettaz, M. J., Paton-Walsh, C., Williams, L. R., Keywood, M. D., and Miljevic, B.: Aging of aerosols emitted from biomass burning in northern Australia, Atmos. Chem. Phys. Discuss., 2016, 1-24, 10.5194/acp-2016-730, 2016.

Shi, Z., Bonneville, S., Krom, M. D., Carslaw, K. S., Jickells, T. D., Baker, A. R., and Benning, L. G.: Iron dissolution kinetics of mineral dust at low pH during simulated atmospheric processing, Atmos. Chem. Phys., 11, 995-1007, 10.5194/acp-11-995-2011, 2011.

Spolaor, A., Vallelonga, P., Gabrieli, J., Cozzi, G., Boutron, C., and Barbante, C.: Determination of Fe2+ and Fe3+ species by FIA-CRC-ICP-MS in Antarctic ice samples, Journal of Analytical Atomic Spectrometry, 27, 310-317, 2012.

Winton, H., Bowie, A., Keywood, M., van der Merwe, P., and Edwards, R.: Suitability of high-volume aerosol samplers for ultra-trace aerosol iron measurements in pristine air masses: blanks, recoveries and bugs, Atmos. Meas. Tech. Discuss., 2016, 1-32, 10.5194/amt-2016-12, 2016.

Winton, V. H. L., Bowie, A. R., Edwards, R., Keywood, M., Townsend, A. T., van der Merwe, P., and Bollhöfer, A.: Fractional iron solubility of atmospheric iron inputs to the Southern Ocean, Marine Chemistry, 177, Part 1, 20-32, http://dx.doi.org/10.1016/j.marchem.2015.06.006, 2015.

Wu, J., Rember, R., and Cahill, C.: Dissolution of aerosol iron in the surface waters of the North Pacific and North Atlantic oceans as determined by a semicontinuous flow-through reactor method, Global Biogeochem. Cycles, 21, GB4010, 10.1029/2006gb002851, 2007.

---

## Referee Report (RR1)

**Review of "Dry season aerosol iron solubility in tropical northern Australia", by V. H. L. Winton, R. Edwards, A. R. Bowie, M. Keywood, A. G. Williams, S. Chambers, P. W. Selleck, M. Desservettaz, M. Mallet, and C. Paton-Walsh (ACP-2016-419).**

The revised paper represents a substantial improvement over the initial submission. However, in my opinion, a number of issues need to be addressed prior to publication.

**Typographical/clarity issues**

| | |
|---|---|
| P4L11 | "attached with" should read "equipped with", or something similar |
| P5L16 | "acidified to" should read "acidified with" |
| P8L10 | "accompanied with" should read "accompanied by" |
| Section 2.3.2 | title should be changed to reflect the fact that the section describes more than total iron concentrations (nss-K, trace elements) |
| P7L5 | "tissuquartz" should be capitalized |
| Section 3.2 | Reference should be made to the relevant figures here.  In addition, the lettering used is confusing:  I see that A and B are taken for dust events, and C presumably refers to coastal conditions, so why are EFGHI used for the biomass burning events (*i.e.* where is D?)? |
| Section 3.3 | This seems more appropriate for the methods section; otherwise, the manuscript skips back and forth between campaign-related information and methods-related information.  In addition, "similar to other studies of instantaneous soluble Fe" should have a reference. |
| P9L15–18 | the Fe data seems to be shown in Figures 3 and 5, not 4 and 6; in addition, it's not clear to me what is meant by "distinct events" |
| P10L4 | parentheses confusing as written |
| Sections 4.1 and 4.2 | it would be clearer if all results were presented using the same units (either of ug/m3 or ng/m3) |
| P10L16 | "topical" should read "tropical" |
| P12L13 | "peaked to" should read "reached" |
| P13L19 | "parties" should read "particles" |
| P14L15 | "courser" should read "coarser" |
| P15L2 | "factional" should read "fractional" |
| P15L19 | "SAFRIED" should read "SAFIRED" |
| P15L29 | "import" should read "important" |
| P16L8 | "Little is known about the fractional Al, Ti and Mn solubility in biomass aerosols future work" should be revised. |
| P16L18 | "desserts" should read "deserts" |
| Figures | consistency needed in the text re: use of "Figure X" vs. "Fig. X" |

**Technical issues**

| | |
|---|---|
| Section 2.3.2 | Do the reported total/soluble Fe values reflect these blank concentrations? It's unclear as written.  What is the variation in the blank values?  Several reported concentrations (GP16, GP15) are not much higher than the blanks, so this requires explanation. |

**Issues with data interpretation / presentation**

Section 4.3       I disagree with the authors' presentation/interpretation of information in Figure 7. First, the data doesn't sit in "two narrow clusters"—the two clusters overlap, and are both quite broad. Second, the total aerosol loading in both clusters is much higher than the Sholkovitz data points, *i.e.* not "moderate". Finally, the reason behind the relationship presented by Sholkovitz (combustion sources with high Fe solubility mix with dust with low solubility) is not the relationship discussed in the present paper, where the solubility was lower when nearby fires were present. A discussion of aged/non-aged combustion Fe would illuminate this discrepancy; as written, it's confusing.

Section 4.6       This section is generally confusing to me—it contains a mixture of raw data, data interpretation, and extrapolation/prediction. I would suggest re-writing in a way that more clearly links the observations in the present paper to the conclusions being made.